# Factors affecting mental health of health care workers during coronavirus disease outbreaks (SARS, MERS & COVID-19): A rapid systematic review

Niels De Brier[1]☯*, Stijn Stroobants[2]☯, Philippe Vandekerckhove[3,4], Emmy De Buck[1,4,5]

1 Centre for Evidence-Based Practice, Belgian Red Cross, Mechelen, Belgium, 2 Humanitarian Services, Belgian Red Cross, Mechelen, Belgium, 3 Belgian Red Cross, Mechelen, Belgium, 4 Department of Public Health and Primary Care, Faculty of Medicine, KU Leuven, Leuven, Belgium, 5 Cochrane First Aid, Mechelen, Belgium

☯ These authors contributed equally to this work.
* Niels.Debrier@rodekruis.be

**Data Availability Statement:** All relevant data are within the manuscript and its Supporting Information files.

## Abstract

### Introduction

The novel Coronavirus Disease (COVID-19) outbreak currently puts health care workers at high risk of both physical and mental health problems. This study aimed to identify the risk and protective factors for mental health outcomes in health care workers during coronavirus epidemics.

### Methods

A rapid systematic review was performed in three databases (March 24, 2020) and a current COVID-19 resource (May 28, 2020). Following study selection, study characteristics and effect measures were tabulated, and data were synthesized by using vote counting. Meta-analysis was not possible because of high variation in risk factors, outcomes and effect measures. Risk of bias of each study was assessed and the certainty of evidence was appraised according to the GRADE methodology.

### Results

Out of 2605 references, 33 observational studies were selected and the identified risk and protective factors were categorized in ten thematic categories. Most of these studies (n = 23) were performed during the SARS outbreak, seven during the current COVID-19 pandemic and three during the MERS outbreak. The level of disease exposure and health fear were significantly associated with worse mental health outcomes. There was evidence that clear communication and support from the organization, social support and personal sense of control are protective factors. The evidence was of very low certainty, because of risk of bias and imprecision.

**Funding:** This work was made possible through funding from the Foundation for Scientific Research of the Belgian Red Cross. All authors are in employment at Belgian Red Cross and receive no other funding. No external funders had a role in study design, data collection and analysis, decision to publish, or preparation of the manuscript.

**Competing interests:** I have read the journal's policy and the authors of this manuscript have the following competing interests: the activities of the Belgian Red Cross include the provision of psychosocial first aid to laypeople. This does not alter our adherence to PLOS ONE policies on sharing data and materials.

## Conclusion

Safeguarding mental health of health care workers during infectious disease outbreaks should not be treated as a separate mental health intervention strategy, but could benefit from a protective approach. This study suggests that embedding mental health support in a safe and efficient working environment which promotes collegial social support and personal sense of control could help to maximize resilience of health care workers. Low quality cross-sectional studies currently provide the best possible evidence, and further research is warranted to confirm causality.

## Introduction

The novel Coronavirus Disease (COVID-19) outbreak, triggered by infection with severe acute respiratory syndrome coronavirus 2 (SARS-CoV-2), has rapidly evolved into a global pandemic [1]. Similar yet less widespread coronavirus epidemics have occurred in the past, notably the severe acute respiratory syndrome (SARS) and Middle East respiratory syndrome (MERS) outbreaks [2, 3]. A common feature of emerging infectious diseases is their rapid increase in incidence, which pressures and potentially exceeds the limits of health care service capacities [4, 5].

The current public health emergency exerts significant physical and mental burden on patients, and also health care workers (HCWs) represent a group at specific risk [6]. Typically, they account for a large percentage of patients in infectious disease outbreaks since they provide care for confirmed or suspected cases and/or generally maintain close contacts while physical distancing is warranted [7, 8]. Moreover, high rates of mental health problems among physicians, nurses and hospital-based personnel during coronavirus disease outbreaks have been frequently described [9, 10]. Mental health problems generally involve a constellation of changes in thinking, feeling and/or behavior, which are deemed undesirable by the person experiencing them and/or by his environment. Such changes can present within a broad range of severity from life's daily hassles to diagnosable psychiatric disorders. Pappa et al. [11] and da Silva [12] provide early evidence that a high proportion of HCWs experience significant levels of anxiety, depression, stress and insomnia during the COVID-19 pandemic and Pan et al. [10] confirmed that the anxiety level of Chinese HCWs significantly increased during the outbreak of COVID-19. Another recent meta-analysis revealed that HCWs exposed to SARS/MERS/COVID-19 reported symptoms of e.g. fear, insomnia, psychological distress, burnout and anxiety [13]. Although these recent meta-analyses clearly indicate high pressure on the mental health of HCWs during the COVID-19 pandemic, personal, social and organizational factors associated with their vulnerability or resilience have not been synthesized.

Considering their great personal risk and pivotal role in tackling this global health crisis, adequate mental health care for HCWs is imperative [14–16]. Although plenty of psychosocial initiatives are described, the most effective strategies in a pandemic context are currently unclear [17, 18]. We performed a rapid systematic review to identify the best possible evidence on risk and protective factors for psychological outcomes in HCWs during coronavirus epidemics. Our results may inform decisions to safeguard HCW's mental health during this and future respiratory infectious disease outbreaks.

## Methods

Because of the timely relevance of the findings to support time-sensitive decisions during the COVID-19 pandemic, we decided to conduct a rapid systematic review. Due to time

constraints inherent to the development of a rapid systematic review, no protocol for the systematic literature searches was registered beforehand with PROSPERO. The reporting of the systematic literature reviews was done according to the steps outlined in Preferred Reporting Items for Systematic Reviews and Meta-Analyses (PRISMA) (see S1 Table) [19].

We conducted a systematic literature search in three databases (MEDLINE, via the PubMed interface, Embase, via the Embase.com interface and PsycINFO, via the APA PsycNET database) and the NIPH systematic and living map on COVID-19 evidence (PubMed results supplemented by regular updates with material retrieved by searches performed by organizations such as World Health Organization, WHO and Centers for Disease Control and Prevention, CDC) [20] to answer the following research question: What are risk factors/protective factors (I) for the mental health (O) of HCWs during a coronavirus disease outbreak, epidemic or pandemic (P)?

Two information specialists independently developed a search strategy based on search terms describing the HCW, the epidemic, and either mental health outcomes or interventions (full search strategies can be found in S1 Text). In the NIPH living COVID-19 map, the retrieved records were filtered on terms related to HCWs within the category of "experiences and perceptions; consequences; social, political, economic aspects", covering studies on mental health. Searches in the databases were ran on March 24, 2020 and in the selected COVID-19 resource on May 28, 2020 to identify the latest studies on COVID-19. Retrieved references from the three databases were imported in Endnote, duplicates were removed, and study selection was performed by one reviewer, and selected studies were critically evaluated by the second reviewer until consensus was reached.

Studies were eligible if they addressed the PICO question and met the following inclusion and exclusion criteria:

- Population: *Included*: studies targeted at all staff which are/were active within a health care setting (e.g. a whole hospital or specific unit, health center, or community health network) during an outbreak of a coronavirus infection, causing the following diseases: SARS, MERS, COVID-19. *Excluded*: studies dealing with other infectious disease outbreaks (e.g. ebola and H1N1 virus).

- Risk or protective factors: *Included*: studies describing any modifiable risk factor or protective factor, which is relevant to take into account when developing either prevention programs or mental health interventions for HCWs in the context of an infectious disease outbreak. Risk factors are here defined as characteristics at organizational, social and personal level that putatively precede and are associated with poor mental health outcomes in HCWs. Protective factors are positive influences that may protect HCWs for developing mental health problems during coronavirus disease outbreaks. Modifiable factors include behaviors, experiences and exposures that may be controlled and changed for maximizing resilience of HCWs during and after these crises. Examples of modifiable factors are: direct contact with patients, dissatisfaction with procedures, changes in work demands, being quarantined as HCW, fear of infection, stigma, vulnerability, clear communication of directives, professional support, social support, perceived self-efficacy, sufficient precautionary measures, training in protection, . . . We adhered to the author's interpretation on the classification of risk/protective factors and outcome measures. *Excluded*: studies describing non-modifiable factors such as gender, age, family history and professional title and factors associated with impact on personal life. Relevant factors were discussed with content experts. Studies solely dealing with prevalence or incidence rates of mental health outcomes are excluded since no data on risk or protective factors can be extracted.

- Outcome: *Included*: any mental health outcome or psychological wellbeing. Mental health problems can present within a broad range of severity. Hence, we not only focused on mental health outcomes reflecting psychiatric symptoms and/or caseness, but also included studies assessing mental health symptomatology reflecting common emotional and social undesirable changes in thinking, feeling and/or behavior. Mental health outcomes included physical symptoms such as pain and fatigue in as far as they were part of mental health questionnaires and thus considered putatively related to psychological suffering. Mental health outcomes were considered in the immediate, short- and long-term as long as association with risk/protective factors directly relevant to the disease outbreak could be examined. *Excluded*: physical health problems related to the inflammatory responses of the human body such as fever, cough, myalgias, chills, headaches, dyspnea, sore throat, nausea/vomiting and diarrhoea.

- Study design: *Included*: (i) the studies of a systematic review if the search strategy and selection criteria were clearly described and if at least three electronic databases were searched; (ii) experimental studies: (quasi or non-)randomized controlled trial (RCT), controlled before and after studies or controlled interrupted time series and (iii) observational studies: cohort and case-control studies, controlled before and after studies and controlled interrupted time series and cross-sectional studies, but measures should have controlled for confounding factors (e.g. matching, multivariate regression analyses). *Excluded*: cross-sectional studies which did not control for confounding factors, case series, letters, qualitative studies, conference abstracts, PhD theses and publications that had not been subject to peer review.

- Other: Only English language studies were included and there were no restrictions regarding publication date or status.

Study characteristics (study type, country, epidemic, population, relevant risk factors for which confounding was taken into account, relevant outcomes with their measurement tool) and study findings [mean difference (MD), adjusted odds ratio (aOR), or regression coefficient (β)] were extracted and tabulated. A p-value < 0.05 was considered statistically significant.

Since meta-analysis was not possible because of a high variety in factors, outcomes and effect measures (adjusted for different confounding factors), we synthesized the data where possible using vote counting based on direction of effect by comparing the number of comparisons showing harm and benefit, regardless of the statistical significance or size of their results [21]. A binomial test (RStudio) was used to assess the significance of evidence for the existence of an effect in either direction. A p-value < 0.05 was considered statistically significant.

Limitations in observational study designs were analyzed by assessing the eligibility criteria, methods for exposure and outcome variables, the strategies for controlling for confounding factors, and follow-up, according to the GRADE (Grading of Recommendations Assessment, Development and Evaluation) criteria. This evaluation was followed by a certainty rating of the body of evidence according to the GRADE methodology [22].

## Results

Out of 2605 references we selected 33 relevant studies (29266 participants) (Fig 1), of which 32 cross-sectional studies, and one uncontrolled before-and-after study. Five studies that fulfilled our predefined eligibility criteria were identified through the specific COVID-19 resource. The majority of the included studies dealt with the mental health of HCWs during the SARS epidemic (n = 23), seven studies during the COVID-19 outbreak in Chinese hospitals and three studies surveyed hospital staff during the MERS epidemic. The studies reported on a wide range of mental health outcomes which could be categorized in eight categories, based on the

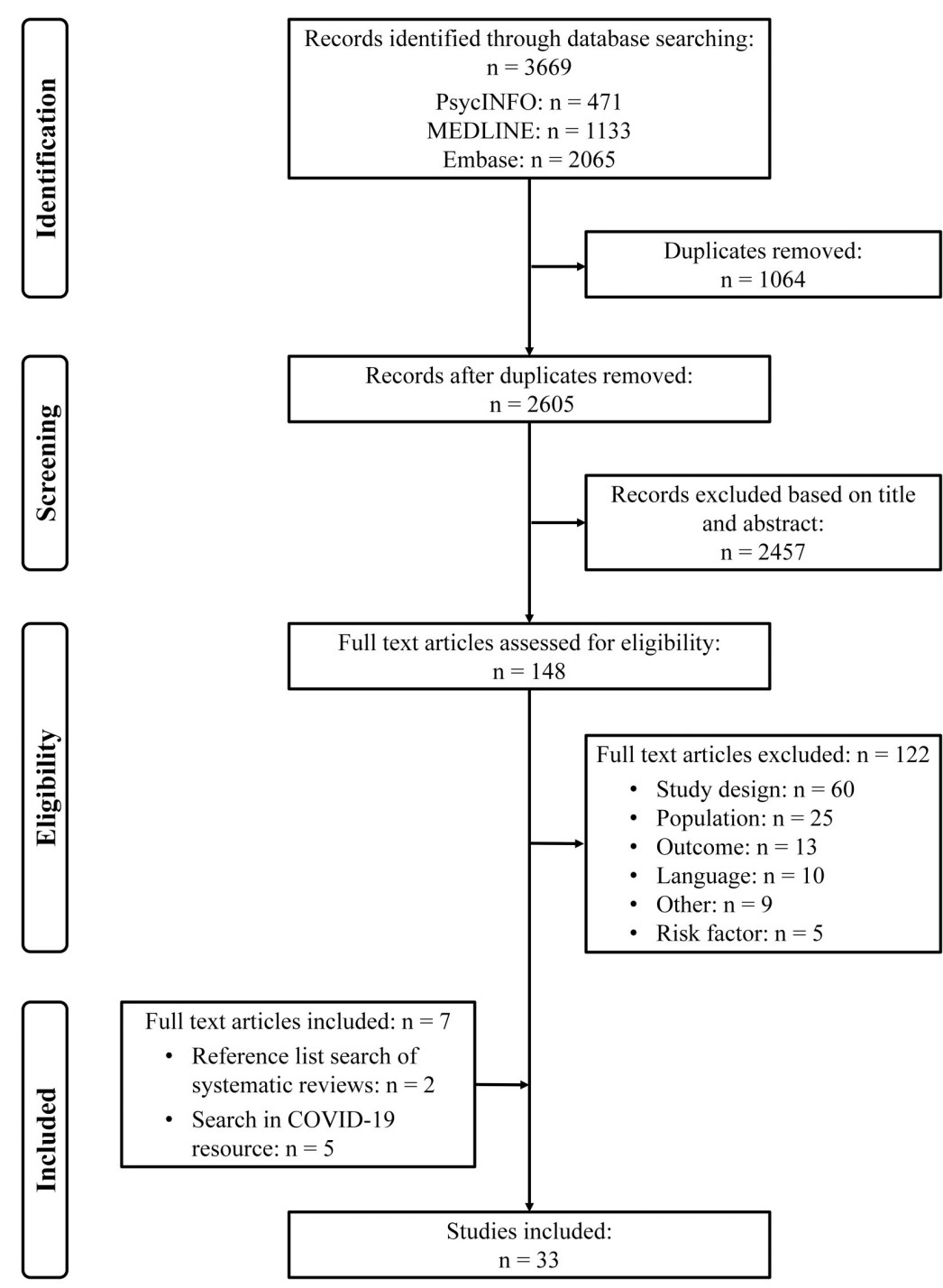

**Fig 1. PRISMA flowchart for the selection of eligible studies.**

measurement scales that were used and detailed in Table 1: acute stress disorder/post-traumatic stress (mainly based on versions of the Impact of Event Scale), anxiety-related symptoms, depression-related symptoms, (perceived) stress, emotional exhaustion and burnout, sleep problems (including insomnia symptoms), anger and general symptoms of psychopathology including nonspecific outcomes such as mental distress, emotional distress,

**Table 1. Study characteristics.** Outcome measures sharing the same number in superscript belong to the same thematic category.

| Author Publication year | Country | Epidemic | Study design | Participants | Outcome measures Measurement tool |
|---|---|---|---|---|---|
| Bai 2004 [23] | Taiwan | SARS | Cross-sectional study | 338 staff members including HCWs and administrative personnel | • Acute stress disorder, according to DSM-IV [1] |
| Chan 2004 [24] | Singapore | SARS | Cross-sectional study | 661 HCWs | • Post-traumatic stress, Impact of Event Scale [1] |
| | | | | | • Psychiatric symptoms, 28-item General Health Questionnaire [8] |
| Chang 2006 [25] | Taiwan | SARS | Cross-sectional study | 211 HCWs | • Emotional exhaustion, measured with a questionnaire developed by the researcher [5] |
| Chen 2005 [26] | Taiwan | SARS | Cross-sectional study | 128 nurses | • Post-traumatic stress, Impact of Event Scale [1] |
| | | | | | • Anxiety, 90-item Symptom Checklist-Revised [2] |
| | | | | | • Depression, 90-item Symptom Checklist-Revised [3] |
| Chen 2006 [27] | Taiwan | SARS | Uncontrolled before-and-after study | 116 volunteers from the nursing staff | • Anxiety, Zung's self-rating anxiety scale [2] |
| | | | | | • Depression, Zung's self-rating depression scale [3] |
| | | | | | • Sleep quality, Pittsburgh sleep quality index [6] |
| Chen 2007 [28] | Taiwan | SARS | Cross-sectional study | 90 HCWs and 82 subjects without contact with SARS patients (e.g. hospital administrators) | • Mental health, part of the general health status, as measured by the Medical Outcome Study Short-Form 36 Survey [8] |
| Chong 2004 [9] | Taiwan | SARS | Cross-sectional study | 1257 staff members including HCWs and administrative workers | • Psychiatric morbidity, 12 item Chinese Health Questionnaire (use of cut-off score) [8] |
| Ho 2005 [29] | Hong Kong | SARS | Cross-sectional study | 97 HCWs who had been infected | • Post-traumatic stress symptoms, The Chinese Impact of Event Scale—Revised [1] |
| Kang 2020 [30] | China | COVID-19 | Cross-sectional study | 994 HCW including medical and nursing staff | • Mental health, based on a cluster analysis according to the scores of the following outcome measures: [8] |
| | | | | | • Post-traumatic stress (symptoms), 22-item Impact of Event Scale |
| | | | | | • Anxiety, 7-item Generalized Anxiety Disorder |
| | | | | | • Depression, 9-item Patient Health Questionnaire |
| | | | | | • Insomnia symptoms, Insomnia Severity Index |
| Kim 2016 [31] | South Korea | MERS | Cross-sectional study | 215 nurses | • MERS-related burnout, Oldenburg Burnout Inventory [5] |
| Koh 2005 [32] | Singapore | SARS | Cross-sectional study | 10511 staff members from 3 SARS and 6 SARS-free hospitals | • Stress at work, questionnaire developed by researchers [4] |
| Lai 2020 [33] | China | COVID-19 | Cross-sectional study | 1257 HCWs | • Depression, 9-item Patient Health Questionnaire [3] |
| | | | | | • Anxiety, 7-item Generalized Anxiety Disorder [2] |
| | | | | | • Insomnia symptoms, Insomnia Severity Index [6] |
| | | | | | • Distress symptoms, 22-item Impact of Event [1] |
| Lancee 2008 [34] | Canada | SARS | Cross-sectional | 587 HCWs | • Psychological disorder, according to DSM-IV [8] |
| Liu 2012 [35] | China | SARS | Cross-sectional study | 549 HCWs | • Depression, Center for Epidemiologic Studies Depression Scale [3] |
| Lu 2020 [36] | China | COVID-19 | Cross-sectional study | 2299 HCWs including 2042 medical staff and 257 administrative staff | • Fear, numeric rating scale [2] |
| | | | | | • Anxiety, Hamilton Anxiety Scale [2] |
| | | | | | • Depression, Hamilton Depression Scale [3] |
| Marjanovic 2007 [37] | Canada | SARS | Cross-sectional study | 333 nurses | • Emotional exhaustion, Maslach Burnout Inventory-General Survey [5] |
| | | | | | • State anger, State-Trait Anger Expression Inventory [7] |
| Maunder 2004 [38] | Canada | SARS | Cross-sectional study | 1557 HCWs | • Psychological stress, Impact of Event scale [1] |

(*Continued*)

**Table 1.** (Continued)

| Author Publication year | Country | Epidemic | Study design | Participants | Outcome measures Measurement tool |
|---|---|---|---|---|---|
| Maunder 2006 [39] | Canada | SARS | Cross-sectional study | 769 HCWs | • Acute stress/post-traumatic stress (symptoms): Post-traumatic stress, Impact of Event Scale [1] |
| | | | | | • General symptoms of psychopathology: Psychological stress, Kessler Psychological Distress Scale [8] |
| | | | | | • Emotional exhaustion & Burnout: Professional burnout, Emotional Exhaustion Scale of the Maslach Burnout Inventory [5] |
| McAlonan 2007 [40] | Hong Kong | SARS | Cross-sectional study | 106 high-risk HCWs and 71 control subjects including non-respiratory medicine workers | • Perceived stress scale [4] |
| | | | | | • Depression, subscale of Depression and Anxiety Scale [3] |
| | | | | | • Anxiety, subscale of Depression and Anxiety Scale [2] |
| | | | | | • Stress, subscale of Depression and Anxiety Scale [4] |
| | | | | | • Post-traumatic stress, Impact of Event Scale [1] |
| Mo 2020 [41] | China | COVID-19 | Cross-sectional study | 180 nurses | • Stress, Chinese version of Stress Overload Scale [4] |
| Nickell 2004 [42] | Canada | SARS | Cross-sectional study | 2001 hospital employees | • Concern for personal of family's health, based on closed and open-ended questions [2] |
| | | | | | • Emotional distress, 12-item General Health Questionnaire [8] |
| Park 2018 [43] | South Korea | MERS | Cross-sectional study | 187 nurses working in a high risk area | • Mental Health, Mental Component Summary of Short Form-36 instrument [8] |
| | | | | | • Perceived stress scale-10 [4] |
| Sim 2004 [44] | Singapore | SARS | Cross-sectional study | 277 HCWs | • Psychiatric morbidity, 28-item General Health Questionnaire (use of cut-off score) [8] |
| | | | | | • Post-traumatic morbidity, Impact of Event Scale [1] |
| Son 2019 [45] | South Korea | MERS | Cross-sectional study | 280 HCWs and administrative staff | • Likelihood of post-traumatic stress symptoms, Korean version of Impact of Event Scale [1] |
| | | | | | • Negative emotional experience, 9-point Likert scale [8] |
| Styra 2008 [46] | Canada | SARS | Cross-sectional study | 248 HCWs from high-risk units | • Post-traumatic stress, Impact of Event Scale [1] |
| Su 2007 [47] | Taiwan | SARS | Cross-sectional study | 70 nurses from SARS units and 32 nurses from non-SARS units | • Depression, Beck Depression Inventory [3] |
| | | | | | • Post-traumatic stress symptoms, Chinese version of Davidson Trauma Scale [1] |
| | | | | | • Presence of insomnia, DSM-IV [6] |
| Tam 2004 [48] | Hong Kong | SARS | Cross-sectional study | 652 frontline HCWs | • Psychiatric morbidity, 12-item Chinese Health Questionnaire (with use of cut-off score) [8] |
| Wong 2005 [49] | Hong Kong | SARS | Cross-sectional study | 466 HCWs | • Mental distress, newly designed scale [8] |
| Wong 2007 [50] | Canada | SARS | Cross-sectional study | 137 and 51 doctors from Hong Kong and Toronto respectively | • Anxiety, Visual Analogue Scale [2] |
| Wu 2009 [6] | China | SARS | Cross-sectional study | 549 HCWs and administrative staff | • Post-traumatic Stress, Impact of Event Scale [1] |
| Xiao 2020 [51] | China | COVID-19 | Cross-sectional study | 180 HCWs | • Anxiety, Self-Rating Anxiety Scale [2] |
| | | | | | • Self-efficacy, General Self-Efficacy Scale [4] |
| | | | | | • Stress, Stanford Acute Stress Reaction Questionnaire [4] |
| | | | | | • Sleep quality, Pittsburgh Sleep Quality Index [6] |
| Zhang 2020 [52] | China | COVID-19 | Cross-sectional study | 1563 medical staff members including frontline workers | • Insomnia symptoms, Insomnia Severity Index [6] |
| Zhu 2020 [53] | China | COVID-19 | Cross-sectional study | 165 HCW (79 doctors, 86 nurses) | • Anxiety, self-rating anxiety scale [2] |
| | | | | | • Depression, self-rating depression scale [3] |

psychological distress, psychological disorder, psychiatric morbidity, psychiatric symptoms, general mental health and negative emotional experience. Acute stress/post-traumatic stress symptoms was the most reported outcome. The study characteristics, including exact outcome measures and scales used, are summarized in Table 1 and S2 Table. Relevant risk factors and protective factors were clustered in six and four thematic categories, respectively, as described below and in S3 Table (detailed summary of findings table)

We identified six categories of risk factors which might be associated with poor mental health in HCW. In first instance, evidence was found that HCWs exposed to coronavirus were at risk of developing mental health problems with 30 of 34 comparisons showing harm (88%, 95%CI [73%;97%], p<0.001) (Table 2) [6, 9, 24, 26, 28, 30, 32, 33, 35–40, 44, 46, 48, 52]. The risk factors included, amongst others, direct contact with patients, working in high risk units, high risk of exposure, and working on the frontline. Three studies reported that HCWs in China being in contact with COVID-19 patients were at significantly higher risk of experiencing symptoms related to post-traumatic stress (aOR: 1.60, 95%CI [1.25;2.04], p<0.001) [33], depression (aOR: 1.52, 95%CI [1.11;2.09], p = 0.01 and aOR: 2.016, 95%CI [1.102;3.685], p = 0.023) [33, 36], anxiety (aOR: 1.57, 95%CI [1.22;2.02], p<0·001 and aOR: 2.062, 95%CI [1.349;3.153], p = 0.001) [33, 36], insomnia (aOR: 2.97, 95%CI [1.92;4.60], p<0.001) [33], fear (aOR: 1.408, 95%CI [1.025;1.933], p = 0.034) [36] and general mental health problems (β: 5.347, 95%CI [3.831;8.184], p<0.001) [30]. A study by Zhang et al. [52] could not demonstrate a significant association between contact with COVID-19 patients and insomnia symptoms (aOR: 1.252, 95%CI [0.960;1.632], p = 0.098). Other studies were conducted during (or after) the SARS epidemic and, when placing a higher value on the largest studies (n>1000), indicate that being exposed (daily) to SARS (aOR: 1.62, 95%CI [1.1;2.4], p = 0.017 and aOR: 1.33, 95% CI [1.19; 1.49], p<0.05) was clearly associated with psychiatric morbidity or more stress at work, respectively [9, 32]. Of note, a study by Maunder et al. [38] could not demonstrate a significant association between contact with SARS patients and post-traumatic symptomatology (β: 0.002, 95%CI [-0.054;0.059], p = 0.95).

Secondly, the included studies (n = 4) suggested that HCWs who were quarantined (100%, 95%CI [48%;100%], p = 0.063) have worse mental health outcomes in all five comparisons (Table 2) [6, 23, 35, 37]. The largest studies (n>500) indicate that being quarantined is

**Table 2. Synthesis of the impact of identified risk and protective factors on the development of mental health problems based on vote counting of the direction of effect.** The unknown directions of effect represent the factors which did not significantly contribute to the multivariate regression analyses and of which the direction of effect was not reported.

| Variable | Direction of effect | | | Vote counting | | |
| --- | --- | --- | --- | --- | --- | --- |
| | Risk (# comparisons) | Protective (# comparisons) | Unknown (# comparisons) | Proportion | 95% CI | P value |
| **Risk factors** | | | | | | |
| Level of disease exposure | 30 | 4 | 12 | 88% | [73;97] | <0.001 |
| Being quarantined | 5 | 0 | 3 | 100% | [48;100] | 0.063 |
| Job stress and dissatisfaction | 9 | 2 | 1 | 82% | [48;98] | 0.065 |
| Risk perception and fear | 12 | 1 | 2 | 92% | [64;100] | 0.003 |
| Stigma | 4 | 0 | 4 | 100% | [40;100] | 0.125 |
| Loss of control and emotional disruption | 3 | 0 | 0 | 100% | [29;100] | 0.250 |
| **Protective factors** | | | | | | |
| Organizational communication and support | 1 | 11 | 2 | 92% | [62;100] | 0.006 |
| Physical safety and training | 3 | 8 | 7 | 73% | [39;94] | 0.227 |
| Social support | 2 | 11 | 3 | 85% | [55;98] | 0.022 |
| Sense of control and coping ability | 0 | 6 | 0 | 100% | [54;100] | 0.031 |

associated with post-traumatic stress (aOR: 2.09, 95%CI [1.00;4.37], p<0.05) or depression (aOR: 5.06, 95%CI [2.12, 12.10], p<0.05) [6, 35]. Of note, quarantine was not identified as a significant risk factor in the univariate analysis, and thus not entered in the regression models of Maunder et al. [39]. Smaller studies (n = 2) also reported significant associations between quarantine and development of acute stress disorder [23] and anger [37]. The latter study could not demonstrate a significant impact of time spent in quarantine on emotional exhaustion [37].

Thirdly, the body of evidence on job stress and dissatisfaction could not demonstrate a clear association with mental health problems but only a suggestive trend, with nine of the 11 comparisons showing harm (82%, 95%CI [48%;98%], p = 0.065) (Table 2) [26, 31, 35, 38, 41, 42, 46, 48, 49]. When focusing on the larger studies (n>500), Maunder et al. [38] found a clear association between job stress and post-traumatic stress (β: 0.208, 95%CI [0.162;0.254], p<0.001), while Liu et al. [35] could not demonstrate a significant association between job stress and depression (aOR: 0.60, 95%CI [0.16;2.19], p>0.05). Tam et al. [48] also reported that job-related stress was significantly associated with psychiatric morbidity (aOR: 4.06, 95% CI [2.28;7.23], p<0.05). Smaller studies (n = 4) found that job stress, impact on work life, working hours per week and being involuntary conscripted to a high risk unit, were significantly associated with burnout, post-traumatic stress, total stress load, or depression and stress reactions, respectively [26, 31, 41, 46]. The latter risk factor was not significantly associated with the presence of symptoms related to anxiety or general mental distress [26, 49]. Moreover, job stress related to precautionary measures or hospital procedures was significantly associated with emotional distress (aOR: 2.9, 95% [1.9;4.6], p<0.05) in a study of Nickell et al. [42], while Maunder et al. [38] found no association between dissatisfaction with procedures and post-traumatic stress.

Fourthly, evidence exists that HCWs experiencing fear or perceiving more risk of becoming infected or infecting others may increasingly suffer from mental health problems with 12 of 13 comparisons showing harm (92%, 95%CI [64%;100%], p = 0.003) (Table 2). Fear of infection and perception of risk were generally significantly associated with post-traumatic stress, anxiety, depression, insomnia, and (mental) distress [6, 29, 35, 38, 42, 45–47, 49, 52], while a significant association between perception of risk to others or family and post-traumatic stress or mental distress could not be demonstrated [46, 49]. Of note, the study by Kim and Choi [31] was not able to demonstrate a significant association between fear for MERS-infection and burnout.

Lastly, although a suggestive trend exists that stigma, including avoidance and social isolation (100%, 95%CI [40%;100%], p = 0.125) [38, 39, 42, 43, 49], and loss of control and emotional disruption (100%, 95%CI [29%;100%], p = 0.250) [29, 45, 49] can be associated with mental distress, no significant relationship could be demonstrated due to the low number of events (Table 2).

Furthermore, four categories of factors which may protect HCWs from developing mental health problems during coronavirus disease outbreaks were identified ("protective factors"). First, 11 out of 12 comparisons (92%, 95%CI [62%;100%], p = 0.006) indicated that HCWs who receive clear communication and support from the organization were less likely to develop mental health problems (Table 2) [24, 27, 30, 37, 46, 48]. The clear communication of directives and precautionary measures was statistically associated with psychiatric symptoms (aOR: 0.51, 95% CI [0.29;0.90], p = 0.020), while a statistical association could not be demonstrated with post-traumatic stress [24]. The provision of support and adequate insurance and compensation by the organization was significantly associated with relieved feelings of anger (β: 0.24, 95%CI [0.13;0.35], p = 0.000) and reduced psychiatric morbidity (aOR: 0.52, 95%CI [0.29;0.93], p<0.05), respectively [37, 48]. Very recently, Kang et al. [30] demonstrated that

HCWs who accessed mental health care services were at lower risk of developing mental health problems (β: -0.868, 95%CI [-1.385;-0.289], p = 0.001) during the COVID-19 outbreak. Other factors such as confidence in the information provided and expressing opinions through staff unions or mass media were not statistically significant in multivariate logistic regression analyses [46, 48]. More in particular, there is evidence that implementing a supportive prevention program, including the availability of a mental health team, training, detailed manpower allocation, and adequate equipment, substantially improved symptoms of anxiety and depression along with sleep quality of the nursing staff [27]. In line with this result, perception of adequate counseling and psychological support from the employer was found to statistically lower the psychiatric morbidity of HCWs (aOR: 0.53, 95%CI [0.31;0.89], p<0.05) [48].

Regarding physical safety and training, the body of evidence could not demonstrate a clear association, but only provided a suggestive trend that provision of physical safety and training protects HCWs from developing mental health problems with eight of the 11 comparisons showing benefit (73%, 95%CI [39%;94%], p = 0.227) (Table 2) [34, 37–39, 42, 46, 48, 50, 52]. Evidence exists that perceived adequacy of training and support was negatively associated with post-traumatic stress (β: -0.22, 95%CI [-0.38;-0.06], p = 0.01), burnout (β: -0.27, 95%CI [-0.44;-0.10], p = 0.002) [39], and psychological disorder (β: -0.20, p = 0.03) [34]. Similarly, trust in precautionary measures, equipment and infection control initiatives also protected HCWs from emotional exhaustion (β: -0.15, 95%CI [-0.26;-0.05], p = 0.005), concern for personal or family health (aOR: 0.4, 95%CI [0.3;0.5], p<0.05) or state anger (β: -0.14, 95%CI [-0.25;-0.03], p = 0.011) [37, 42]. Moreover, HCWs who are uncertain regarding effective disease control during the current COVID-19 pandemic might be at high risk of facing insomnia symptoms (aOR: 3.297, 95%CI [1.284;8.469], p = 0.013) [52]. On the contrary, Wong et al. [50] found that training in handling infectious disease outbreaks was significantly associated with increased anxiety amongst HCW (aOR: 5.41, 95%CI [1.02;28.79], p = 0.048). No statistical associations between doubt about protection, unprotected exposure, infection control measures/guidelines, infection prevention training and protective facilities and mental health outcomes could be demonstrated [38, 39, 46, 48, 52].

Next, HCWs perceiving social support are more resilient against mental distress with 11 of 13 comparisons showing benefit (85%, 95%CI [55%;98%], p = 0.022) (Table 2) [24, 25, 31, 48, 51]. Of particular interest, Xiao et al. [51] found that levels of social support for HCWs who recently treated patients with COVID-19 infection, were significantly associated with self-efficacy (β: 0.023, 95%CI [0.011;0.035], p<0.001), and negatively associated with degree of anxiety (β: –0.781, 95%CI [-0.948;-0.614], p<0.0001) and stress (β: -0.704, 95%CI [-1.161;-0.247], p = 0.003). A clear association with sleep quality could not be demonstrated. There is evidence that especially the support from supervisors and colleagues is beneficially associated with post-traumatic stress (aOR: 0.33, 95% CI [0.16;0.69], p = 0.003) and psychiatric symptoms (aOR: 0.35, 95% CI [0.17;0.69], p = 0.003), while a significant effect could not be demonstrated for support from family or the ability to talk about concerns [24, 48]. Other studies (n = 2) reported social support measures to be related to significantly lower levels of emotional exhaustion or burnout [25, 31]. Four other social support factors such as sense of coherence, interaction and appreciation by the community did not statistically contribute to regression model analyses.

The included studies (n = 4) indicated that factors related to sense of control and coping ability are associated with HCWs resilience against the development of mental health problems in all six comparisons (100%, 95%CI [54%;100%], p = 0.031) (Table 2) [29, 37, 43, 45]. Levels of vigor and hardiness were negatively associated with perceived stress, emotional exhaustion, mental health, and state anger [37, 43]. Significant associations of perceived self-efficacy or coping ability in HCWs could not be demonstrated [29, 45]. Specific coping strategies were

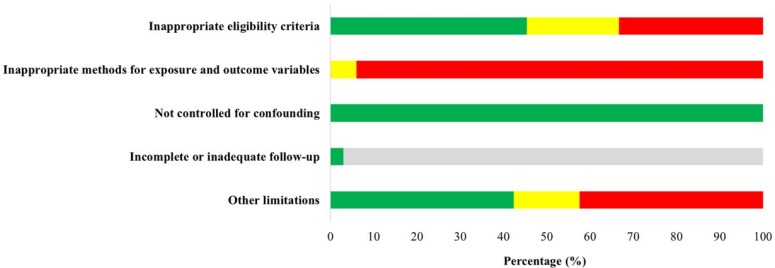

**Fig 2. Risk of bias graph showing each item presented as percentages across all observational studies with green: Low risk of bias, yellow: Unclear risk of bias, red: High risk of bias and grey: Not applicable.**

not included in our data synthesis, but it was shown that venting, humor, and altruistic acceptance were coping strategies related to a statistically significant decrease of post-traumatic stress or morbidity or symptoms of depression during the SARS outbreak [6, 35, 44], while for other strategies this association could not be demonstrated [44]. Very recently, Zhu et al. [53] indicated that positive coping by HCWs during the COVID-19 pandemic was significantly associated with lower risk of developing depression- or anxiety-related symptoms.

Finally, we assessed the limitations in study design for all included studies individually (Fig 2 and S4 Table). The 33 studies included were observational studies, leading to an initial "low" certainty level. The certainty of evidence was downgraded (-1) for limitations in study design. All but one study had a cross-sectional design, and no causal relationships can be inferred. We assessed the studies at high risk of recall (mostly studies dealing with MERS and SARS epidemic) and selection bias, with about 20% of the studies having a response rate lower than 50%. The response rate varied between 10.5 and 94.9%. The overall certainty of evidence was further downgraded because of imprecision (-1) due to limited sample sizes, large variability of the results and/or lack of data, resulting in "very low"-certainty evidence meaning that any estimate of effect is very uncertain.

## Discussion

We identified 32 cross-sectional studies and 1 before-and-after study. Most of the studies were performed during the SARS outbreak (n = 23), while seven very recent studies were executed during the current COVID-19 pandemic and three during the MERS outbreak. A relatively large body of evidence resulted in the following risk factors: level of disease exposure and health fear. A limited number of studies also suggested being quarantined as HCW, job stress and dissatisfaction, and stigma as risk factors. We also identified factors that are protective for mental health problems, including clear communication and support from the organization, social support, and sense of control and coping ability. Less compelling evidence was found about providing physical safety and training to HCWs as protective factors. We included seven studies on COVID-19 that fulfilled our selection criteria but since the pandemic is still raging, other studies will probably come available the coming months and years. The results of this rapid review will not only serve as a knowledge base for supporting the HCWs during the current acute outbreak but will also allow hospital-based managers and decision-makers to prepare prevention programs for a second wave or other infectious disease outbreaks in the future.

This systematic review has several limitations. First, because of time pressure of making timely relevant results available during the COVID-19 pandemic, no protocol was registered with PROSPERO, study selection and data extraction was done by only one reviewer and only English language studies were included. The latter has resulted in the exclusion of some

studies, particularly from Asian countries. However, we were able to detect 33 individual studies against the predefined eligibility criteria of which about 80% were conducted in Asia. We only screened a limited number of databases. Moreover, relevant bibliographies and clinical trial registries were not searched. On the other hand, we used a sensitive search strategy, checked a very comprehensive COVID-19 resource and two reviewers independently compared the included studies against the selection criteria. A formal assessment of publication bias was not possible due to large variation in outcome measures and effect sizes. Second, the quality of the included studies was very low, as the majority of studies had a cross-sectional design, which made it difficult to infer causal relationships from the results. However, within the available cross-sectional studies, we decided to only focus on those studies that controlled for confounding factors. Of note, the included studies did not involve the occupational mental health of HCWs prior to the disease outbreak as a control measure. Recent systematic reviews, identified through an abbreviated literature search and screening, clearly showed that especially occupational stress and burnout are already highly prevalent (even as high as 80%) among medical doctors and nurses in the pre-pandemic workplace [54–57]. Estimates are however complicated due to variability in definitions and assessment methods and differences between regions and occupational roles [54, 57]. A cross-sectional study prior to the current pandemic in Spain showed that, although burnout is a major problem among nurses working in emergency departments, the average level of perceived stress was found within a normal range [58]. The study by McAlonan et al. [40] showed that the perceived stress levels in both high- and low-risk HCWs during the SARS outbreak were higher than the normative value of a US community sample. Furthermore, two (meta-analytic) prevalence studies performed during the current COVID-19 pandemic showed that levels of anxiety and depression of the HCWs during the outbreak in China were significantly higher than that of the HCWs during the non-outbreak period or the national norm [10, 59]. Third, since there was a high variation in effect measures, risk factors, and measurement tools, it was difficult to synthesize the findings, and we used vote counting based on direction of effect as synthesis method. As a consequence, statistical heterogeneity could not be assessed. To avoid fragmentizing the body of evidence, we combined several mental health outcomes in this analysis, for the different categories of risk/protective factors. Vote counting provides no information on magnitude of effect and takes no account of the differences in relative sizes of each study. Since the mental health outcomes were grouped and the risk and protective factors were clustered in ten thematic categories, this synthesis method is hence not a nuanced approach. Indeed, these thematic categories (in)directly involve multiple affiliated factors and, by way of example, witnessing multiple deaths could be related to 'level of disease exposure' and risk of transmission to family was presently categorized under 'risk perception and fear'. Nonetheless the results of the individual studies were tabulated and extensively discussed.

The findings of this systematic review are highly relevant for supporting the HCWs facing the current COVID-19 pandemic and advanced the conclusions of previous systematic evaluations. One review (search date 2009) dealt with the relation of the occurrence of work-related critical incidents and mental health of HCWs, and confirmed that treating victims of terror or disease outbreaks or, more general, treating patients in critical care treatments, had a small to medium impact on post-traumatic stress symptoms, anxiety, and depression in HCWs compared with a control, non-exposed group [60]. However, this review did not identify risk or protective factors determining the mental health status of HCWs, and the population was not limited to disease outbreaks. Very recent meta-analyses confirmed that HCWs exposed to (SARS, MERS or) COVID-19 suffered from a wide range of mental health problems but no risk or protective factors were identified [10–13]. Another systematic review [61] studied the psychological wellbeing of HCWs involved in the SARS epidemic (search date 2015). The data

were synthesized in a thematic narrative analysis. The review concluded that the mental health impact of SARS on HCWs was associated with occupational role, training/preparedness, high-risk work environments, quarantine, role-related stressors, perceived risk, social support, social rejection/isolation and impact of SARS on personal or professional life. In addition to the evidence from the SARS epidemic, the current systematic review now also includes more recent evidence from other coronavirus epidemics, such as MERS and COVID-19. An added value of the current study is the focus on studies that used methods to take into account confounding factors, and the use of quantitative data extraction and synthesis for listing which factors are significantly associated with worse or better mental health outcomes. The body of evidence identified in this study indicates that especially the provision of support (programs) by health care organizations, social support (by colleagues) and personal sense of control and coping ability putatively benefit the mental health of HCW. At the same time, the review quantitatively substantiated that high-risk work environments and health fear put HCWs at risk of mental health problems.

Although the reviewers searched for specific mental health interventions for supporting HCWs during coronavirus disease outbreaks, there was little if any evidence for such interventions. However, several risk and protective factors for mental health of HCWs on the organizational, social and personal level were identified. The available evidence in the context of coronavirus disease outbreaks therefore suggests that safeguarding their mental health should not be treated as a separate mental health intervention strategy, but requires an integrated protective approach. Embedding access to mental health support in a safe and efficient working environment which promotes collegial social support and personal sense of control could help to maximize resilience of health workers during the current global health crisis. Our review might have several implications for practice and we formulated a number of considerations and recommendations:

- Sense of support by the organization: Apart from structural support and efficient communication, monitoring of well-being of HCWs is recommended. Monitor from a viewpoint of supportive concern. Awareness by HCWs of internal (e.g. a mental health support team) and external mental health initiatives (e.g. mental health hotlines or websites, contact information of mental health organizations) should be ensured. Actively refer HCWs to such initiatives in case of concern.

- Provide opportunity to talk, listen to concerns and offer empathic support: these guidelines protect mental health in general and empathic and supportive interactions between colleagues as well as with supervisors should be promoted. Providing specific occasions to share concerns and support could be helpful. Training such supportive skills could provide further benefit.

- Protect physical safety: providing a safe working environment and adequate and timely information about the required precautionary measures, could help to correct overly strong risk perceptions if applicable. In general, promote consulting trustworthy sources of information about the epidemic. Adequate formal training could further protect both physical and mental health.

- Reduce the impact of changing job demands (unfamiliar tasks, changing working conditions, work overload, . . .): this could include clear definitions and communication of changed duties, and safeguarding sufficient rest is available. Considerations for the COVID-19 outbreak by the WHO [62] for example suggest to initiate, encourage and monitor work breaks.

- Maximize sense of control of HCWs: the aforementioned provision of information, clear communication and efficient organization could partially contribute in this respect. Normalize stress reactions and praise achievements. Promote the idea of challenge instead of threat in light of the outbreak. Paying attention to feelings and cognitions of control could be beneficial in resilience training programs.

- HCWs that are quarantined should receive continuous support from supervisors and colleagues. Quarantine should not last longer than necessary. Of note, attention should also be paid to HCWs who are self-quarantined to avoid infecting family members.

- Provide additional attention to HCWs in high-risk occupations. The above recommendations might be specifically beneficial for this subgroup, which proves vulnerable to developing mental health problems.

Our review also highlights the need for future research, with a focus on better study designs. These could include proper control group and longitudinal designs, with a follow-up measurement after the crisis period. In addition, studies defining a specific exposed and non-exposed group (cohort-type studies) or a case and control group (case-control type studies), where both groups are properly matched based on important confounding variables, would provide useful information. Since we only identified one study that measured the impact of a specific program, including mental health aspects, there is a clear need for studies reporting on the effectiveness of concrete mental health interventions.

Lastly, the multiple uncertainties about the COVID-19 outbreak and the rapidly growing research make it necessary to provide the scientific community with high-quality and timely updates of the relevant evidence. As this review deals with time sensitive results, we screened for additional relevant experimental or observational studies in the NIPH living COVID-19 map during the publication process (search date Oct 26). In this time period, no high-quality studies at low risk of bias which could substantially impact on the review's conclusions were identified. Eight additional cross-sectional studies, which suffer from coexisting methodological issues, fulfilled the eligibility criteria. These studies did not reveal any new risk factors and further underpin the findings of this review. When briefly discussing the risk factors, studies by Cai et al. [63] and Wankowicz et al. [64] confirmed that the rate of mental health problems is significantly increased in frontline HCWs. Further, there is growing evidence that risk perception and health fear are independent factors for developing mental distress [65, 66] while Sarboozi Hoseinabadi et al. [67] could not demonstrate a significant association between fear of COVID-19 infection and burnout symptomatology. It was also confirmed that loneliness because of social isolation as well as job stress are associated with increased mental distress or burn-out symptomatology, respectively [66, 67]. Regarding the protective factors, further evidence shows that receiving adequate information and availability of protective measures are significantly associated with reduced severity of mental health problems [66, 68, 69]. Sarboozi Hoseinabadi et al. [67] could not demonstrate a significant association between social support from family and friends or hospital resources and COVID-19-related burnout. Finally, Huang et al. [70] found that problem-coping style is the common influencing factor for anxiety in nurses. It is of note that the studies of Arpacioglu et al. [65] and García-Fernández et al. [68] included HCWs as well as non-HCWs and results between groups could not be distinguished.

In conclusion, several organizational factors, including clear communication and directives, and organizational support seem paramount in addition to protective factors such as HCWs perceptions of coping ability and social support. Future efforts should especially target HCWs working on the frontline. Our results may inform decisions to safeguard the mental health of HCWs during this and future respiratory infectious disease outbreaks. Due to the high

variation in outcome measures, risk factors, and measurement tools, it was not possible to perform a meta-analysis but the data were synthesized by using vote counting based on direction of effect. Although it is difficult to infer causal relationships from the evidence, cross-sectional studies currently provide the best possible evidence for developing practical recommendations in this context. High-quality controlled studies are needed for establishing casual relationships and identifying the most effective interventions.

## Supporting information

**S1 Table. PRISMA checklist.**
(DOCX)

**S2 Table. Detailed study characteristics.**
(DOCX)

**S3 Table. Summary of findings.**
(DOCX)

**S4 Table. Risk of bias assessment.**
(DOCX)

**S1 Text. Search strategies.**
(DOCX)

## Acknowledgments

We thank Koen Van Praet (Psychosocial Intervention Service, Belgian Red Cross) for useful input to our findings.

## Author Contributions

**Conceptualization:** Stijn Stroobants, Philippe Vandekerckhove, Emmy De Buck.

**Data curation:** Niels De Brier, Stijn Stroobants, Emmy De Buck.

**Formal analysis:** Niels De Brier, Emmy De Buck.

**Funding acquisition:** Philippe Vandekerckhove.

**Investigation:** Niels De Brier, Stijn Stroobants, Emmy De Buck.

**Methodology:** Niels De Brier, Emmy De Buck.

**Supervision:** Philippe Vandekerckhove, Emmy De Buck.

**Validation:** Stijn Stroobants, Philippe Vandekerckhove.

**Writing – original draft:** Niels De Brier, Emmy De Buck.

**Writing – review & editing:** Stijn Stroobants, Philippe Vandekerckhove.

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
