## [Decision Letter · Decision Letter 0]

6 May 2020

PONE-D-20-11380

Factors affecting mental health of health care workers during coronavirus disease outbreaks: a rapid systematic review

PLOS ONE

Dear Dr. De Brier,

Thank you for submitting your manuscript to PLOS ONE. After careful consideration, we feel that it has merit but does not fully meet PLOS ONE’s publication criteria as it currently stands. Therefore, we invite you to submit a revised version of the manuscript that addresses the points raised during the review process.

We would appreciate receiving your revised manuscript by Jun 20 2020 11:59PM. To enhance the reproducibility of your results, we recommend that if applicable you deposit your laboratory protocols in protocols.io, where a protocol can be assigned its own identifier (DOI) such that it can be cited independently in the future. For instructions see: http://journals.plos.org/plosone/s/submission-guidelines#loc-laboratory-protocols

We look forward to receiving your revised manuscript.

Kind regards,

Andrew Carl Miller

Academic Editor

PLOS ONE

Journal Requirements:

"I have read the journal's policy and the authors of this manuscript have the following

competing interests: the activities of the Belgian Red Cross include the provision of

psychosocial first aid to laypeople."

Reviewers' comments:

Reviewer's Responses to Questions

**Comments to the Author**

1. Is the manuscript technically sound, and do the data support the conclusions?

Reviewer #1: Yes

Reviewer #2: Yes

2. Has the statistical analysis been performed appropriately and rigorously? 

Reviewer #1: Yes

Reviewer #2: No

3. Have the authors made all data underlying the findings in their manuscript fully available?

Reviewer #1: Yes

Reviewer #2: Yes

4. Is the manuscript presented in an intelligible fashion and written in standard English?

Reviewer #1: Yes

Reviewer #2: Yes

5. Review Comments to the Author

Reviewer #1: Thank you for the opportunity to review this timely and well-written manuscript.

- Please state explicitly that the systematic review followed the steps outlined in Preferred Reporting Items for Systematic Reviews and Meta-Analyses (PRISMA).

- Why was the project not registered with PROSPERO? I suspect due to time constraints and desire to publish the findings to aid with the pandemic, but please report the reason.

- Was a librarian involved in developing the search strategies?

- Were relevant bibliographies also searched?

- Were searches limited by date or publication status?

- Were clinical trial registries also searched to limit publication bias, including: ClinicalTrials.gov, World Health Organization International Clinical Trials Registry Platform (WHO ICTRP), and the Australian New Zealand Clinical Trials Registry (ANZCTR), etc.?

- Inclusion / Exclusion criteria need to be more completely described.

- The exact outcomes need to be more explicitly described. For example, it is not sufficient to state: Outcomes were categorized 67 in 8 categories, based on the measurement scales that were used. Risk and protective factors were categorized in 68 6 and 4 thematic categories, respectively. One should state the exact variables and scales used. Provide references for the validated scales.

- It is good that they used the GRADE system for grading evidence.

- A visual representation of risk-of-bias assessments (e.g. RoB graph) would be helpful.

- Please report risk of publication bias using the Egger or Begg-Mazumdar methods (or other validated method).

- Please describe method used to determine heterogeneity, and what level of heterogeneity (e.g. I2 ≥50%, ≥75%, etc.) was the threshold beyond which data would not be suitable to combine into a meta-analysis.

Page 4, Line 80: please list subject # in parentheses behind “28 studies”

- Results: when listing results for an aggregate number of studies (e.g. 4 included studies …) please list the (n) in parentheses.

- Discussion: Many US physicians have self-imposed a form of quarantine from family (only going to work, living in garage, office, car, etc.) so as not to risk exposing them.

- Is information available about the effects of healthcare workers being displaced from home / family.

- Line 269: Does the evidence suggest a most-effective mechanism for employers to check in / monitor the mental health of their front-line healthcare workers?

Limitations

- Not registered with PROSPERO

- Search strategy not developed by librarian

- Restricted only to English language

- Many important databases left out of search strategy (many of which could have been searched in English).

- Determination of inclusion/exclusion performed by 1 person.

- Data extraction performed by 1 person.

- Needs risk-of-bias graph.

- Risk of publication bias not reported.

- Heterogeneity not reported.

Reviewer #2: You are to be commended for performing the review in such a short time.

Search: You did not mention languages searched. Which did you search?

Results:

You identified 23 studies about SARS, 3 about MERS and 2 Chinese studies about Covid-19. The fact that there are only two Covid studies should appear in your title and be emphasised in your abstract and conclusions

You have identified these risk factors; direct contact, high risk units, high exposure risk, and working on the front line

You have identified these protective factors: clear communication, social support, sense of control, and coping ability, and less about physical safety and training.

In the media the health care workers and populations around the world have emphasised also the slowness of authorities to respond to the crisis, slow and inadequate social distancing and quarantining, the absence of personal protective equipment, witnessing multiple deaths, risks of transmission to family members, and long shifts and fatigue. There are thus some differences between these and the outcomes you found.

Do you know an expert who has done media searches and has the software ready or could you put out a call who could identify the key words associated with Covid 19 to provide more evidence about Covid-19? and run a separate search including terms for health care workers.

You are right to emphasise the very low study quality due to variation in risk factors, outcomes and measuring tools and that needs to be strongly emphasised in your abstract and conclusions.

Another key problem is that there are no baseline measures pre-pandemic for any workers and you need to stress that. A search for systematic reviews of healthcare workers stress and depression could provide you with a baseline.

6. PLOS authors have the option to publish the peer review history of their article (what does this mean?). If published, this will include your full peer review and any attached files.

Reviewer #1: Yes: Andrew C. Miller

Reviewer #2: Yes: Roger E. Thomas

---

## [Author Response · Author response to Decision Letter 0]

13 May 2020

Reply to the remarks of the reviewers of manuscript ID PONE-D-20-11380

Dear Dr. Miller

Dear Editor

Thank you for the work done on manuscript ID PONE-D-20-11380 entitled "Factors affecting mental health of health care workers during coronavirus disease outbreaks: a rapid systematic review" by Niels De Brier, Stijn Stroobants, Philippe Vandekerckhove and Emmy De Buck.

Reviewers commented positively on our submission and had some comments and suggestions. We now have revised the manuscript and believe that we have upgraded its quality. Below, we detail how we have addressed all of the individual points raised. We hope that the revisions made and responses given are clear such that the manuscript will now be acceptable for publication. 

Yours sincerely,

Niels De Brier, corresponding author on behalf of all co-authors. 

 

Journal requirements 

Please ensure that your manuscript meets PLOS ONE's style requirements, including those for file naming

We doublechecked the journal guidelines and ensured that the manuscript meets the PLOS ONE's style requirements. 

Thank you for stating the following in the Competing Interests section: "I have read the journal's policy and the authors of this manuscript have the following competing interests: the activities of the Belgian Red Cross include the provision of psychosocial first aid to laypeople."

Please confirm that this does not alter your adherence to all PLOS ONE policies on sharing data and materials, by including the following statement: "This does not alter our adherence to PLOS ONE policies on sharing data and materials.” Please include your updated Competing Interests statement in your cover letter; we will change the online submission form on your behalf.

In response, we updated the Competing Interest statement in the cover letter as follows: “I have read the journal's policy and the authors of this manuscript have the following competing interests: the activities of the Belgian Red Cross include the provision of psychosocial first aid to laypeople. This does not alter our adherence to PLOS ONE policies on sharing data and materials.”

Reviewer #1

Thank you for the opportunity to review this timely and well-written manuscript.

We thank the reviewer for his positive comment and for providing clear guidance on how to upgrade the quality of the manuscript. 

Please state explicitly that the systematic review followed the steps outlined in Preferred Reporting Items for Systematic Reviews and Meta-Analyses (PRISMA).

In response, the sentence in lines 52-53 now reads as follows: “The reporting of the systematic literature reviews was done according to the steps outlined in Preferred Reporting Items for Systematic Reviews and Meta-Analyses (PRISMA) (see S1 Table)”. 

Why was the project not registered with PROSPERO? I suspect due to time constraints and desire to publish the findings to aid with the pandemic, but please report the reason.

To address this comment, the sentence in lines 49-52 now reads as follows: “Because of the timely relevance of the findings to support time-sensitive decisions during the COVID-19 pandemic, we decided to conduct a rapid systematic review. Due to time constraints inherent to the development of a rapid systematic review, no protocol for the systematic literature searches was registered beforehand with PROSPERO.”

Was a librarian involved in developing the search strategies?

The search strategies were developed by review authors NDB and EDB who are trained information specialists and the sentence in lines 60-61 now read as follows: “Two information specialists independently developed a search strategy based on search terms describing the HCW, the epidemic, and either mental health outcomes or interventions (full search strategies can be found in S2 Text).”

Were relevant bibliographies also searched?

No bibliographies were searched due to time constraints. We now acknowledge this limitation in the manuscript in line 243: “Moreover, relevant bibliographies and clinical trial registries were not searched.”

Were searches limited by date or publication status?

There were no constraints in selection criteria regarding publication date or status. In response, the sentence in lines 82-83 now reads as follows: “Other: Only English language studies were included and there were no restrictions regarding publication date or status.”

Were clinical trial registries also searched to limit publication bias, including: ClinicalTrials.gov, World Health Organization International Clinical Trials Registry Platform (WHO ICTRP), and the Australian New Zealand Clinical Trials Registry (ANZCTR), etc.?

No clinical trial registries were searched due to time constraints. We now acknowledge this limitation in the manuscript in line 243: “Moreover, relevant bibliographies and clinical trial registries were not searched.”

Inclusion / Exclusion criteria need to be more completely described.

To address this comment, we have now substantially revised the description of the inclusion and exclusion criteria in lines 65-83 as follows:

“Studies were eligible if they addressed the PICO question and met the following inclusion and exclusion criteria:

• Population: Included: studies with HCWs in the (pre-)hospital setting during an outbreak of a coronavirus infection, causing the following diseases: SARS, MERS, COVID-19. Excluded: studies dealing with other infectious disease outbreaks (e.g. ebola and H1N1 virus). 

• Risk or protective factors: Included: studies describing any risk factor or protective factor, which is relevant to take into account when developing either prevention programs or mental health interventions for HCWs. Excluded: studies describing non-modifiable factors such as gender, age and professional title and factors associated with impact on personal life. Relevant factors were discussed with content experts.

• Outcome: Included: any mental health outcome or psychological wellbeing. 

• Study design: Included: (i) the studies of a systematic review if the search strategy and selection criteria were clearly described and if at least three electronic databases or relevant bibliographies were searched; (ii) experimental studies: (quasi or non-)randomized controlled trial (RCT), controlled before and after studies or controlled interrupted time series and (iii) observational studies: cohort and case-control studies, controlled before and after studies and controlled interrupted time series and cross-sectional studies, but measures should have controlled for confounding factors (e.g. matching, multivariate regression analyses). Excluded: cross-sectional studies which did not control for confounding factors, case series, letters, qualitative studies, conference abstracts and PhD theses. 

• Other: Only English language studies were included and there were no restrictions regarding publication date or status.” 

The exact outcomes need to be more explicitly described. For example, it is not sufficient to state: Outcomes were categorized 67 in 8 categories, based on the measurement scales that were used. Risk and protective factors were categorized in 68 6 and 4 thematic categories, respectively. One should state the exact variables and scales used. Provide references for the validated scales.

We agree with the reviewer and have now described the outcomes more explicitly in the Results section as outlined below. The references of the exact outcome measures and scales used are provided in Table 1 and S3 and S4 Table. 

Lines 103-110: “The studies reported on a wide range of mental health outcomes which could be categorized in eight categories, based on the measurement scales that were used and detailed in Table 1: acute stress disorder/post-traumatic stress (mainly based on versions of the Impact of Event Scale), anxiety-related symptoms, depression-related symptoms, (perceived) stress, emotional exhaustion and burnout, sleep problems (including insomnia symptoms), anger and general symptoms of psychopathology including nonspecific outcomes mental distress, emotional distress, psychological distress, psychological disorder psychiatric morbidity, psychiatric symptoms, general mental health and negative emotional experience. “

Moreover, the sentence in lines 110-111 now reads as follows: “The study characteristics, including exact outcome measures and scales used, are summarized in Table 1 and S3 Table.”

Finally, we added numbers to Table 1 to transparently highlight the link between the outcome measure/scale used and the thematic category. The Table caption now reads as follows: “Table 1. Study characteristics. Outcome measures sharing the same number in superscript belong to the same thematic category.”

It is good that they used the GRADE system for grading evidence.

We thank the reviewer for his positive comment. 

A visual representation of risk-of-bias assessments (e.g. RoB graph) would be helpful.

In response, we made a risk of bias graph as Fig. 2 and the caption reads as follows: ”Risk of bias graph showing each item presented as percentages across all observational studies with green: low risk of bias, yellow: unclear risk of bias, red: high risk of bias and grey: not applicable. 

Please report risk of publication bias using the Egger or Begg-Mazumdar methods (or other validated method).

Based on the Cochrane and GRADE handbook, tests for funnel plot asymmetry should be used only when there are at least 10 individual studies included in the meta-analysis, because when there are fewer studies the power of the tests is too low to distinguish chance from real asymmetry. Besides the fact that we were not able to perform meta-analyses due to different effect sizes, there were less than 10 studies for each outcome per factor. 

To address this comment and highlight this study limitation, the sentence in lines 245-246 now reads as follows: “A formal assessment of publication bias was not possible due to large variation in outcome measures and effect sizes.”

Please describe method used to determine heterogeneity, and what level of heterogeneity (e.g. I2 ≥50%, ≥75%, etc.) was the threshold beyond which data would not be suitable to combine into a meta-analysis.

As highlighted in lines 9-10, 88-89 and 331-333, a meta-analysis was not possible because of a high variety in risk factors, outcomes and effect measures (adjusted for different confounding factors) and we synthesized the data where possible using vote counting based on direction of effect by comparing the number of comparisons showing harm and benefit, regardless of the statistical significance or size of their results. As a consequence, we were not able to perform I2-statistics. 

In response, we highlighted this limitation in lines 253-255: “Third, since there was a high variation in outcome measures, risk factors, and measurement tools, it was difficult to synthesize the findings, and we used vote counting based on direction of effect as synthesis method. As a consequence, statistical heterogeneity could not be assessed.”

Page 4, Line 80: please list subject # in parentheses behind “28 studies”

In response, the sentence in lines 99-100 now reads as follows: “Out of 2605 references we selected 28 relevant studies (22523 participants) (Fig. 1), of which 27 cross-sectional studies, and one uncontrolled before-and-after study.”

Results: when listing results for an aggregate number of studies (e.g. 4 included studies …) please list the (n) in parentheses.

This comment is readily taken care of. 

Discussion: Many US physicians have self-imposed a form of quarantine from family (only going to work, living in garage, office, car, etc.) so as not to risk exposing them. 

In response, we added this discussion to lines 314-315: “Of note, attention should also be paid to HCWs who are self-quarantined to avoid infecting family members.” 

Is information available about the effects of healthcare workers being displaced from home / family.

Factors related to the personal life of the HCWs were not included in this review. To address this comment, the sentence in lines 71-72 reads as follows: “Excluded: studies describing non-modifiable factors such as gender, age and professional title and factors associated with impact on personal life.”

Line 269: Does the evidence suggest a most-effective mechanism for employers to check in / monitor the mental health of their front-line healthcare workers?

The evidence we gathered does not put forward a most effective monitoring strategy in the context of infectious disease outbreaks. We recommend monitoring of health care workers based upon their increased risk of developing mental health problems. Considering the significance of support factors, we would suggest that any form of monitoring is executed from a viewpoint of supportive concern (not a viewpoint of maladjustment detection), putatively promoting sense of support in HCWs. In response, we added the following sentence in line 293: “Monitor from a viewpoint of supportive concern.”

Limitations

- Not registered with PROSPERO

- Search strategy not developed by librarian

- Restricted only to English language

- Many important databases left out of search strategy (many of which could have been searched in English).

- Determination of inclusion/exclusion performed by 1 person.

- Data extraction performed by 1 person.

- Needs risk-of-bias graph.

- Risk of publication bias not reported.

- Heterogeneity not reported.

In response, we incorporated the individual points raised in the limitation section in the discussion in lines 240-261 as follows:

“This systematic review has several limitations. First, because of time pressure of making timely relevant results available during the COVID-19 pandemic, no protocol was registered with PROSPERO, study selection and data extraction was done by only one reviewer and only English language studies were included. We only screened a limited number of databases. Moreover, relevant bibliographies and clinical trial registries were not searched. On the other hand, we used a sensitive search strategy, surveilled relevant grey literature, and two reviewers independently compared the included studies against the selection criteria. A formal assessment of publication bias was not possible due to large variation in outcome measures and effect sizes. Second, the quality of the included studies was very low, as the majority of studies had a cross-sectional design, which made it difficult to infer causal relationships from the results. However, within the available cross-sectional studies, we decided to only focus on those studies that controlled for confounding factors. Of note, the included studies did not involve the occupational mental health of HCWs prior to the disease outbreak as a control measure. Recent systematic reviews identified through an abbreviated literature search and screening clearly showed that especially occupational stress and burnout are already highly prevalent (even as high as 80%) among medical doctors and nurses in the pre-pandemic workplace [50-52]. Third, since there was a high variation in outcome measures, risk factors, and measurement tools, it was difficult to synthesize the findings, and we used vote counting based on direction of effect as synthesis method. As a consequence, statistical heterogeneity could not be assessed. To avoid fragmentizing the body of evidence, we combined several mental health outcomes in this analysis, for the different categories of risk/protective factors. Vote counting provides no information on magnitude of effect and takes no account of the differences in relative sizes of each study. Since the risk and protective factors were clustered in ten thematic categories, these categories (in)directly involve multiple affiliated factors and, hence, this synthesis method is not a nuanced approach. By way of example, witnessing multiple deaths is related to ‘level of disease exposure’ and risks of transmission to family present was categorized under ‘risk perception and fear’.”

Reviewer #2

You are to be commended for performing the review in such a short time

We thank the reviewer for his/her positive comments.

Search: You did not mention languages searched. Which did you search?

In response, the sentence in lines 82-83 now reads as follows: “Other: Only English language studies were included and there were no restrictions regarding publication date or status.”

Results:

You identified 23 studies about SARS, 3 about MERS and 2 Chinese studies about Covid-19. The fact that there are only two Covid studies should appear in your title and be emphasised in your abstract and conclusions

We agree with the reviewer and added this information now to the abstract, the summary section of the discussion and conclusion as outlined below. Since we were already transparent about the characteristics of the included studies throughout the result section (lines 101-103, Table 1 and Table S3) and now emphasized this information also in the abstract, discussion and conclusion, we prefer to only highlight the selection criteria regarding population, i.e. coronavirus infections, in the title. 

We added the following sentences to 

lines 14-15: “Most of these studies (n=23) were performed during the SARS outbreak, three during the MERS outbreak and two during the current COVID-19 pandemic.“

lines 224-226: “Most of the studies were performed during the SARS outbreak (n=23) while three studies during the MERS outbreak and two very recent studies during the current COVID-19 pandemic.”

lines 232-234: “We only included two studies on COVID-19 that fulfilled our selection criteria, which is probably due to our search date on March 24, which is still quite early in the pandemic. Other studies will probably come available the coming months.”

And lines 336-337: “Only two studies on COVID-19 that fulfilled the eligibility criteria were identified but the body of evidence is expected to grow the coming months.”

You have identified these risk factors; direct contact, high risk units, high exposure risk, and working on the front line

You have identified these protective factors: clear communication, social support, sense of control, and coping ability, and less about physical safety and training.

In the media the health care workers and populations around the world have emphasised also the slowness of authorities to respond to the crisis, slow and inadequate social distancing and quarantining, the absence of personal protective equipment, witnessing multiple deaths, risks of transmission to family members, and long shifts and fatigue. There are thus some differences between these and the outcomes you found. Do you know an expert who has done media searches and has the software ready or could you put out a call who could identify the key words associated with Covid 19 to provide more evidence about Covid-19? and run a separate search including terms for health care workers.

We find it difficult to integrate this suggestion, because of the methodological rigourness of our systematic review. Although findings reported in the media could be timely relevant, they are mostly based on expert opinion, anecdotical evidence or case studies. These study types do not meet our predefined eligibility criteria, stating that studies should have taken measures to control for confounding factors. News reports extracted from peer-reviewed publications (which controlled for confounding factors) should be identified with the sensitive search strategy and included in this manuscript. Moreover, we already summarized the findings from the excluded studies on COVID-19 in the discussion on lines 234-239. 

The risk factors highlighted by the reviewer are implicitly in line with factors identified in our review. We now elaborated on the scope of the identified risk factors as follows in lines 258-261: “Since the risk and protective factors were clustered in ten thematic categories, these categories (in)directly involve multiple affiliated factors and, hence, this synthesis method is not a nuanced approach. By way of example, witnessing multiple deaths is related to ‘level of disease exposure’ and risks of transmission to family present was categorized under ‘risk perception and fear’”. 

Moreover, while the evidence on mental health and psychosocial needs of patients, family members, HCWs, … during the COVID-19 outbreak is rapidly emerging, it is more important to update the findings of this review when new evidence becomes available rather than including too low certainty evidence from the media. In response, the sentences in lines 232-234 now read as follows: “We only included two studies on COVID-19 that fulfilled our selection criteria, which is probably due to our search date on March 24, which is still quite early in the pandemic. Other studies will probably come available the coming months.”

And in lines 324-326: “Lastly, the multiple uncertainties about the COVID-19 outbreak and the rapidly growing research make it necessary to provide the scientific community with high-quality and timely updates of the relevant evidence.”

You are right to emphasise the very low study quality due to variation in risk factors, outcomes and measuring tools and that needs to be strongly emphasised in your abstract and conclusions.

In response, we added following sentences to the abstract and conclusion

in lines 9-10: “Meta-analysis was not possible because of high variation in risk factors, outcomes and effect measures.”

in lines 23-25: “Low quality cross-sectional studies currently provide the best possible evidence, and further research is warranted to confirm causality.”

And in lines 331-336: “Due to the high variation in outcome measures, risk factors, and measurement tools, it was not possible to perform a meta-analysis but the data were synthesized by using vote counting based on direction of effect. Although it is difficult to infer causal relationships from the evidence, cross-sectional studies currently provide the best possible evidence for developing practical recommendations. High-quality controlled studies are needed for establishing casual relationships and identifying the most effective interventions.”

Another key problem is that there are no baseline measures pre-pandemic for any workers and you need to stress that. A search for systematic reviews of healthcare workers stress and depression could provide you with a baseline.

We thank the reviewer for his comment and we agree that adequate pre-pandemic control groups are lacking in the included studies. To address this comment, we elaborated on mental health of HCW prior to the COVID-19 outbreak in lines 249-253: “Of note, the included studies did not involve the occupational mental health of HCWs prior to the disease outbreak as a control measure. Recent systematic reviews identified through an abbreviated literature search and screening clearly showed that especially occupational stress and burnout are already highly prevalent (even as high as 80%) among medical doctors and nurses in the pre-pandemic workplace [50-52].”

Moreover, the sentence in lines 318-319 now reads as follows: “These could include proper control groups and longitudinal designs, with a follow-up measurement after the crisis period.”

Finally, we identified one systematic review with our search strategies dealing with critical incidents such as infectious disease outbreaks and terroristic attacks on incidence of post-traumatic stress symptoms, anxiety, and depression in exposed versus non-exposed health professionals (baseline measure). The sentence in lines 263-267 now reads as follows: “One review (search date 2009) dealt with the relation of the occurrence of work-related critical incidents and mental health of HCWs, and confirmed that treating victims of terror or disease outbreaks or, more general, treating patients in critical care treatments, had a small to medium impact on post-traumatic stress symptoms, anxiety, and depression in HCWs compared with a control, non-exposed group [53]”.

---

## [Decision Letter · Decision Letter 1]

25 May 2020

PONE-D-20-11380R1

Factors affecting mental health of health care workers during coronavirus disease outbreaks: a rapid systematic review

PLOS ONE

Dear Dr. De Brier,

Thank you for submitting your manuscript to PLOS ONE. After careful consideration, we feel that it has merit but does not fully meet PLOS ONE’s publication criteria as it currently stands. Therefore, we invite you to submit a revised version of the manuscript that addresses the points raised during the review process.

We look forward to receiving your revised manuscript.

Kind regards,

Andrew Carl Miller

Academic Editor

PLOS ONE

Reviewers' comments:

Reviewer's Responses to Questions

**Comments to the Author**

1. If the authors have adequately addressed your comments raised in a previous round of review and you feel that this manuscript is now acceptable for publication, you may indicate that here to bypass the “Comments to the Author” section, enter your conflict of interest statement in the “Confidential to Editor” section, and submit your "Accept" recommendation.

Reviewer #1: All comments have been addressed

Reviewer #2: (No Response)

2. Is the manuscript technically sound, and do the data support the conclusions?

Reviewer #1: Partly

Reviewer #2: Yes

3. Has the statistical analysis been performed appropriately and rigorously? 

Reviewer #1: Yes

Reviewer #2: Yes

4. Have the authors made all data underlying the findings in their manuscript fully available?

Reviewer #1: Yes

Reviewer #2: No

5. Is the manuscript presented in an intelligible fashion and written in standard English?

Reviewer #1: Yes

Reviewer #2: Yes

6. Review Comments to the Author

Reviewer #1: - Please indent the first line of new paragraphs.

- Line 33, spelling: outbreaks

- The limitations of the search methodology resulted in a large number of missed studies -- this really undermines the message and solid writing. Cutting out new (and often Asian) literature is a major limitation. Much of the new literature is not on pubmed yet. Some are available as pre-prints. Also, many studies from regions that encountered the virus first (e.g. China) are published in other (non-PubMed) databases (e.g. China National Knowledge Infrastructure (CHKD-CNKI); information/Chinese Scientific Journals database (CSJD-VIP); LILACS, etc.). Even common databases like SCOPUS and Web of Science were not searched. Within a few minutes of searching I was able to identify many missed articles. A partial list includes:

---- Guo J, Liao L, Wang B, et al. Psychological Effects of COVID-19 on Hospital Staff: A National Cross-Sectional Survey of China Mainland. SSRN Electron J 2020; published online March 24. DOI:10.2139/ssrn.3550050.

---- Huang JZ, Han MF, Luo TD, Ren AK, Zhou XP. [Mental health survey of 230 medical staff in a tertiary infectious disease hospital for COVID-19]. Zhonghua Lao Dong Wei Sheng Zhi Ye Bing Za Zhi 2020; 38: E001.

---- Liu C, Yang Y, Zhang XM, Xu X, Dou Q-L, Zhang W-W. The prevalence and influencing factors for anxiety in medical workers fighting COVID-19 in China: A cross-sectional survey. medRxiv 2020; : 2020.03.05.20032003.

---- Liu Z, Han B, Jiang R, et al. Mental Health Status of Doctors and Nurses During COVID-19 Epidemic in China. SSRN Electron J 2020; published online March 24. DOI:10.2139/ssrn.3551329.

---- Zhu Z, Xu S, Wang H, et al. COVID-19 in Wuhan: Immediate Psychological Impact on 5062 Health Workers. medRxiv 2020; : 2020.02.20.20025338.

---- Zhang C, Yang L, Liu S, et al. Survey of Insomnia and Related Social Psychological Factors Among Medical Staff Involved in the 2019 Novel Coronavirus Disease Outbreak. Front Psychiatry 2020; 11: 306.

---- Tan BYQ, Chew NWS, Lee GKH, et al. Psychological Impact of the COVID-19 Pandemic on Health Care Workers in Singapore. Ann Intern Med 2020. DOI:10.7326/M20-1083

---- Lu W, Wang H, Lin Y, Li L. Psychological status of medical workforce during the COVID-19 pandemic: A cross-sectional study. Psychiatry Res 2020; 288: 112936. PMID: 32276196

---- Du J, Dong L, Wang T, et al. Psychological symptoms among frontline healthcare workers during COVID-19 outbreak in Wuhan. Gen Hosp Psychiatry 2020; published online April. DOI:10.1016/j.genhosppsych.2020.03.011.

---- A survey of psychological status and sleep quality of nursing interns during the outbreak of COVID-19 infection. J South Med Univ. 2020;3:346-350.

---- Research on the Model of Psychological Support and Tutoring in New Coronarvirus Pneumonia. Healthcare Res Pract. 2020:2:6-10.

---- Psychological problems and intervention of first-line medical staff in the new coronavirus pneumonia isolation area. J Xinjiang Med Univ. 2020;4:386-390.

---- Study on the status quo and correlation between stress response and sympathetic fatigue of clinical first-line medical staff fighting new coronavirus pneumonia. Pract J Cardiovasc Pulm Dis. 2020;2:9-12.

Reviewer #2: 1. It is important that the title reflects that you analysed only two Covid 19 studies. Readers then know what to expect and can look elsewhere for more Covid studies in later publications or if you research an update.

[This is the response you made: "You identified 23 studies about SARS, 3 about MERS and 2 Chinese studies about Covid-19. The fact that there are only two Covid studies should appear in your title and be emphasised in your abstract and conclusions

We agree with the reviewer and added this information now to the abstract, the summary section of the discussion and conclusion as outlined below. Since we were already transparent about the characteristics of the included studies throughout the result section (lines 101-103, Table 1 and Table S3) and now emphasized this information also in the abstract, discussion and conclusion, we prefer to only highlight the selection criteria regarding population, i.e. coronavirus infections, in the title.

We added the following sentences to

lines 14-15: “Most of these studies (n=23) were performed during the SARS outbreak, three during the MERS outbreak and two during the current COVID-19 pandemic.“

lines 224-226: “Most of the studies were performed during the SARS outbreak (n=23) while three studies during the MERS outbreak and two very recent studies during the current COVID-19 pandemic.”

lines 232-234: “We only included two studies on COVID-19 that fulfilled our selection criteria, which is probably due to our search date on March 24, which is still quite early in the pandemic. Other studies will probably come available the coming months.”

And lines 336-337: “Only two studies on COVID-19 that fulfilled the eligibility criteria were identified but the body of evidence is expected to grow the coming months.”

2. You have no baseline measures. It is important from your literature search that you provide a baseline about stress levels of health care workers in non-pandemic situations.

This is the rssponse you made:"Another key problem is that there are no baseline measures pre-pandemic for any workers and you need to stress that. A search for systematic reviews of healthcare workers stress and depression could provide you with a baseline.

We thank the reviewer for his comment and we agree that adequate pre-pandemic control groups are lacking in the included studies. To address this comment, we elaborated on mental health of HCW prior to the COVID-19 outbreak in lines 249-253: “Of note, the included studies did not involve the occupational mental health of HCWs prior to the disease outbreak as a control measure. Recent systematic reviews identified through an abbreviated literature search and screening clearly showed that especially occupational stress and burnout are already highly prevalent (even as high as 80%) among medical doctors and nurses in the pre-pandemic workplace [50-52].”

7. PLOS authors have the option to publish the peer review history of their article (what does this mean?). If published, this will include your full peer review and any attached files.

Reviewer #1: Yes: Andrew C. Miller

Reviewer #2: Yes: Roger E. Thomas

---

## [Author Response · Author response to Decision Letter 1]

3 Jun 2020

Reply to the remarks of the reviewers of manuscript ID PONE-D-20-11380R1

Dear Dr. Miller

Dear Editor

Thank you for the work done on manuscript ID PONE-D-20-11380R1 entitled "Factors affecting mental health of health care workers during coronavirus disease outbreaks: a rapid systematic review" by Niels De Brier, Stijn Stroobants, Philippe Vandekerckhove and Emmy De Buck.

Reviewers had additional comments and suggestions to improve the manuscript. We now have updated the study selection by including a specific COVID-19 resource, profoundly revised the manuscript and believe that we have upgraded its quality. Below, we detail how we have addressed all of the individual points raised. We hope that the revisions made and responses given are clear such that the manuscript will now be acceptable for publication. 

Yours sincerely,

Niels De Brier, corresponding author on behalf of all co-authors. 

 

Reviewer #1

Please indent the first line of new paragraphs.

In response, all first lines of new paragraphs in the manuscript are now indented. 

Line 33, spelling: outbreaks

This comment is readily taken care of. The sentence in lines 31-33 now reads as follows: “Similar yet less widespread coronavirus epidemics have occurred in the past, notably the severe acute respiratory syndrome (SARS) and Middle East respiratory syndrome (MERS) outbreaks [2,3].”

The limitations of the search methodology resulted in a large number of missed studies -- this really undermines the message and solid writing. Cutting out new (and often Asian) literature is a major limitation. Much of the new literature is not on pubmed yet. Some are available as pre-prints. Also, many studies from regions that encountered the virus first (e.g. China) are published in other (non-PubMed) databases (e.g. China National Knowledge Infrastructure (CHKD-CNKI); information/Chinese Scientific Journals database (CSJD-VIP); LILACS, etc.). Even common databases like SCOPUS and Web of Science were not searched. 

Within a few minutes of searching I was able to identify many missed articles. A partial list includes:

Guo J, Liao L, Wang B, et al. Psychological Effects of COVID-19 on Hospital Staff: A National Cross-Sectional Survey of China Mainland. SSRN Electron J 2020; published online March 24. DOI:10.2139/ssrn.3550050.

Huang JZ, Han MF, Luo TD, Ren AK, Zhou XP. [Mental health survey of 230 medical staff in a tertiary infectious disease hospital for COVID-19]. Zhonghua Lao Dong Wei Sheng Zhi Ye Bing Za Zhi 2020; 38: E001.

Liu C, Yang Y, Zhang XM, Xu X, Dou Q-L, Zhang W-W. The prevalence and influencing factors for anxiety in medical workers fighting COVID-19 in China: A cross-sectional survey. medRxiv 2020; : 2020.03.05.20032003.

Liu Z, Han B, Jiang R, et al. Mental Health Status of Doctors and Nurses During COVID-19 Epidemic in China. SSRN Electron J 2020; published online March 24. DOI:10.2139/ssrn.3551329.

Zhu Z, Xu S, Wang H, et al. COVID-19 in Wuhan: Immediate Psychological Impact on 5062 Health Workers. medRxiv 2020; : 2020.02.20.20025338.

Zhang C, Yang L, Liu S, et al. Survey of Insomnia and Related Social Psychological Factors Among Medical Staff Involved in the 2019 Novel Coronavirus Disease Outbreak. Front Psychiatry 2020; 11: 306.

Tan BYQ, Chew NWS, Lee GKH, et al. Psychological Impact of the COVID-19 Pandemic on Health Care Workers in Singapore. Ann Intern Med 2020. DOI:10.7326/M20-1083

Lu W, Wang H, Lin Y, Li L. Psychological status of medical workforce during the COVID-19 pandemic: A cross-sectional study. Psychiatry Res 2020; 288: 112936. PMID: 32276196

Du J, Dong L, Wang T, et al. Psychological symptoms among frontline healthcare workers during COVID-19 outbreak in Wuhan. Gen Hosp Psychiatry 2020; published online April. DOI:10.1016/j.genhosppsych.2020.03.011.

A survey of psychological status and sleep quality of nursing interns during the outbreak of COVID-19 infection. J South Med Univ. 2020;3:346-350.

Research on the Model of Psychological Support and Tutoring in New Coronarvirus Pneumonia. Healthcare Res Pract. 2020:2:6-10.

Psychological problems and intervention of first-line medical staff in the new coronavirus pneumonia isolation area. J Xinjiang Med Univ. 2020;4:386-390.

Study on the status quo and correlation between stress response and sympathetic fatigue of clinical first-line medical staff fighting new coronavirus pneumonia. Pract J Cardiovasc Pulm Dis. 2020;2:9-12.

We cannot agree more with the reviewer. Initially, we had only included two studies on COVID-19 which is mainly due to our search date on March 24 (which is still quite early in the pandemic) and also due to our strict eligibility criteria (excluding studies that did not control for confounding, or did not include modifiable risk factors). New studies concerning COVID-19 and mental health have become available the past two months, and most of the abovementioned studies were published after our search date (and thus not picked up by our original search).

To address this comment and identify recently published studies on COVID-19, we now have searched the NIPH systematic and living map on COVID-19 evidence(https://www.nornesk.no/forskningskart/NIPH_mainMap.html). In this resource, PubMed results are supplemented by regular updates with material retrieved by searches performed by organizations such as WHO and CDC, and we could specifically filter studies with health care workers and focusing on aspects including mental health. In this way we identified five additional studies that met our eligibility criteria. Two of these studies (Lu et al. and Zhang et al.) were also listed by the reviewer above. The majority of the other studies listed above were now also identified with the additional search, but were excluded because they did not fulfill our selection criteria, e.g. either the study was still in preprint phase and not peer reviewed (e.g. Guo et al., Liu et al., Zhu et al.), no risk/protective factors were measured or the factors measured were non-modifiable, such as demographic factors (e.g. Tan et al.), the study did not fulfill the study design criteria and/or no measures were taken to control for confounding (e.g. Du et al.) or the study was published in Chinese (e.g. Huang et al.).

More studies will become available the coming months and years and the rapidly growing research make it necessary to provide the scientific community with high-quality and timely updates of this rapid review. The ultimate goal of this rapid review is to timely provide a knowledge base for supporting the HCWs during the current acute outbreak but also to allow hospital-based managers and decision-makers to prepare prevention programs for a second wave or other infectious disease outbreaks in the future. 

In total, 33 observational studies were now selected with seven during the current COVID-19 pandemic Following study selection, the data of the new studies were extracted and tabulated (Table 1, Table S3-S5) and included in the data synthesis (Table 2). The new findings are also narratively synthesized in the results section. The results of Figures 1 and 2 are updated as well. All changes can be found in the marked-up copy of our manuscript. 

Reviewer #2

It is important that the title reflects that you analysed only two Covid 19 studies. Readers then know what to expect and can look elsewhere for more Covid studies in later publications or if you research an update.

We have identified five additional studies on COVID-19 that met our eligibility criteria trough a search in the NIPH living COVID-19 map. Since we now included seven very recently published studies on COVID-19, we believe it is not appropriate to emphasize the number of studies in the title but, in response to the reviewer’s comment and to be completely transparent, we now explicitly added the coronavirus diseases which are included in the review. The title now reads as: “Factors affecting mental health of health care workers during coronavirus disease outbreaks (SARS, MERS & COVID-19): a rapid systematic review”

You have no baseline measures. It is important from your literature search that you provide a baseline about stress levels of health care workers in non-pandemic situations.

We thank the reviewer for his comment and now further elaborated on non-pandemic mental health measures in HCWs, citing additional references (lines 261-271): “Recent systematic reviews identified through an abbreviated literature search and screening clearly showed that occupational stress and burnout can be highly prevalent (even as high as 80%) among medical doctors and nurses in the pre-pandemic workplace [55-58]. Estimates are however complicated due to variability in definitions and assessment methods and differences between regions and occupational roles [55,58]. A cross-sectional study prior to the current pandemic in Spain showed that, although burnout is a major problem among nurses working in emergency departments, the average level of perceived stress was found within a normal range [59]. The study by McAlonan et al. [39] showed that the perceived stress levels in both high- and low-risk HCWs during the SARS outbreak were higher than the normative value of a US community sample. Furthermore, two (meta-analytic) prevalence studies performed during the current COVID-19 pandemic showed that levels of anxiety and depression of the HCWs during the outbreak in China were significantly higher than that of the HCWs during the non-outbreak period or the national norm [10,60].“

Of note, the objective of the review was to identify any risk of protective factor which is relevant to take into account when developing either prevention programs or mental health interventions for HCWs and not to map and compare prevalence rates of mental health problems (and thus studies only reporting incidences of mental health problems were excluded).

---

## [Decision Letter · Decision Letter 2]

26 Aug 2020

PONE-D-20-11380R2

Factors affecting mental health of health care workers during coronavirus disease outbreaks (SARS, MERS & COVID-19): a rapid systematic review

PLOS ONE

Dear Dr. de Brier,

Thank you for submitting your manuscript to PLOS ONE. After careful consideration, we feel that it has merit but does not fully meet PLOS ONE’s publication criteria as it currently stands. Therefore, we invite you to submit a revised version of the manuscript that addresses the points raised during the review process.

Please address the definitional and methodological queries posed by the reviewers.

We look forward to receiving your revised manuscript.

Kind regards,

Rosemary Frey

Academic Editor

PLOS ONE

Reviewers' comments:

Reviewer's Responses to Questions

**Comments to the Author**

1. If the authors have adequately addressed your comments raised in a previous round of review and you feel that this manuscript is now acceptable for publication, you may indicate that here to bypass the “Comments to the Author” section, enter your conflict of interest statement in the “Confidential to Editor” section, and submit your "Accept" recommendation.

Reviewer #2: All comments have been addressed

Reviewer #3: (No Response)

2. Is the manuscript technically sound, and do the data support the conclusions?

Reviewer #2: Yes

Reviewer #3: Partly

3. Has the statistical analysis been performed appropriately and rigorously? 

Reviewer #2: Yes

Reviewer #3: No

4. Have the authors made all data underlying the findings in their manuscript fully available?

Reviewer #2: No

Reviewer #3: Yes

5. Is the manuscript presented in an intelligible fashion and written in standard English?

Reviewer #2: Yes

Reviewer #3: Yes

6. Review Comments to the Author

Reviewer #2: The topic has resulted in many studies ad as the discussion between the reviewer and the authors shows there are important differences of opinion.

My recommendation is that the authors review the new studies found by the reviewer and apply their criteria and be clear which can be included, and then I suggest that the editor adjudicates. This is then fair to readers and medical staff.

Reviewer #3: The authors in this systematic review attempted to identify the risk and protective factors for mental health outcomes in health care workers during coronavirus epidemics. The topic is important, pertinent and timely. However, I have some concerns about this study:

1- I appreciate short and concise introductions, particularly in meta-analysis, and to a lesser extent in systematic reviews. However, the current state of knowledge in the field is not sufficiently detailed. In order to identify the risk and protective factors for coronavirus-related mental health outcomes, those outcomes need to be characterized and described.

2- PROSPERO is prioritizing COVID-related submissions and are registering protocols within days. The registration would have been recommendable, not only to avoid to a certain extent post-hoc decisions, but because they provide useful recommendations. In this case, I believe they would have asked the authors to better describe the PICO framework.

3- The literature search is inefficient and under my point of view probably the main reason why relevant hits were lost along the process. Not sure the addition of keywords as “Zika virus’/exp”, “‘Zika fever’/exp”, “hotline:ab,ti” led to the inclusion of many relevant hits. However, demanding studies to include one of the keywords in the third paragraph in the literature search in order to be included, must have limited the detection capacity of relevant hits.

4- I am not convinced by the definition of population regarding exposure to coronavirus. Several studies evaluated HCW, particularly during SARS pandemic, both during outbreak and after the outbreak. It seem like authors considered posttraumatic stress symptoms as well, which may appear months/years after. Also, how was hospital setting defined? Were GPs or HCW working in the community excluded?

5- The definition of a risk or a protective factor is really ambiguous and this is particularly problematic. This way it is not possible to replicate what the authors have done. For instance, looking at the studies in the following systematic review (PMID: 32658823) that were not included in this systematic review, it is difficult to know if it was an strict definition of risk/protective factor (or outcome), an inefficient literature search, or losses of studies during the screening process, which led to the authors missing studies that seem relevant to me.

6- Further explanation of which outcomes were included and excluded would be advisable. How were physical and mental health outcomes defined and differentiated?

7- Pooling together any risk/protective factor with any mental health outcome to provide quantitative results seems questionable to me. Also, do not understand why the outcomes are too heterogenous for a meta-analysis, but not for vote counting..

8- I do not agree with the decision of considering p-value < 0.10 significant regardless of the method and this decision should be better justified.

9- Including only studies in English in this topic is a limitation as many studies come from Asia.

7. PLOS authors have the option to publish the peer review history of their article (what does this mean?). If published, this will include your full peer review and any attached files.

Reviewer #2: **Yes: **Roger E. Thomas.

Reviewer #3: No

---

## [Author Response · Author response to Decision Letter 2]

18 Sep 2020

Reply to the remarks of the reviewers of manuscript ID PONE-D-20-11380R2

Dear Dr. Frey

Dear Editor

Thank you for the work done on manuscript ID PONE-D-20-11380R2 entitled "Factors affecting mental health of health care workers during coronavirus disease outbreaks (SARS, MERS & COVID-19): a rapid systematic review" by Niels De Brier, Stijn Stroobants, Philippe Vandekerckhove and Emmy De Buck.

Reviewers had additional comments and suggestions to improve the manuscript. We now have revised the manuscript and believe that we have upgraded its quality. Below, we detail how we have addressed all of the individual points raised. We hope that the revisions made and responses given are clear such that the manuscript will now be acceptable for publication. 

Yours sincerely,

Niels De Brier, corresponding author on behalf of all co-authors. 

Reviewer #1

Reviewer #1 had no further comments on the revised manuscript. 

Reviewer #2

The topic has resulted in many studies and as the discussion between the reviewer and the authors shows there are important differences of opinion. My recommendation is that the authors review the new studies found by the reviewer and apply their criteria and be clear which can be included, and then I suggest that the editor adjudicates. This is then fair to readers and medical staff.

We profoundly elaborated on this comment in the previous round of revision. As recommended by Reviewer #1 previously, we updated the search on May 28 and we included five additional studies on COVID-19 that met our eligibility criteria. The initial search date was on March 24 which was still quite early in the pandemic and the newly found studies were published after our search date (and thus not picked up by our original search).

Reviewer #3

The authors in this systematic review attempted to identify the risk and protective factors for mental health outcomes in health care workers during coronavirus epidemics. The topic is important, pertinent and timely. However, I have some concerns about this study:

We thank the reviewer for highlighting the timely relevance of this rapid systematic review. 

1- I appreciate short and concise introductions, particularly in meta-analysis, and to a lesser extent in systematic reviews. However, the current state of knowledge in the field is not sufficiently detailed. In order to identify the risk and protective factors for coronavirus-related mental health outcomes, those outcomes need to be characterized and described.

In response, we further detailed the emerging state of knowledge in the introduction and also included the description of the mental health outcomes in the introduction in lines 39-50 as follows: “Moreover, high rates of mental health problems among physicians, nurses and hospital-based personnel during coronavirus disease outbreaks have been frequently described [9,10]. Mental health problems generally involve a constellation of changes in thinking, feeling and/or behavior, which are deemed undesirable by the person experiencing them and/or by his environment. Such changes can present within a broad range of severity from life’s daily hassles to diagnosable psychiatric disorders. Pappa et al. [11] and da Silva [12] provide early evidence that a high proportion of HCWs experience significant levels of anxiety, depression, stress and insomnia during the COVID-19 pandemic and Pan et al. [10] confirmed that the anxiety level of Chinese HCWs significantly increased during the outbreak of COVID-19. Another recent meta-analysis revealed that HCWs exposed to SARS/MERS/COVID-19 reported symptoms of e.g. fear, insomnia, psychological distress, burnout and anxiety [13]. Although these recent meta-analyses clearly indicate high pressure on the mental health of HCWs during the COVID-19 pandemic, personal, social and organizational factors associated with their vulnerability or resilience have not been synthesized.”

2- PROSPERO is prioritizing COVID-related submissions and are registering protocols within days. The registration would have been recommendable, not only to avoid to a certain extent post-hoc decisions, but because they provide useful recommendations. In this case, I believe they would have asked the authors to better describe the PICO framework.

To scientifically support the Belgian Red Cross’ response to the early stages of the COVID-19 pandemic, the review was developed in a very short time frame between March 24 and April 7 with an update on May 28. Due to the given time constraints, no protocol was published beforehand. We acknowledged the limitation of this decision in lines 285-287 as follows: “This systematic review has several limitations. First, because of time pressure of making timely relevant results available during the COVID-19 pandemic, no protocol was registered with PROSPERO, study selection and data extraction was done by only one reviewer and only English language studies were included.”

To better describe the PICO framework, thereby taking into account the comments outlined below, we further elaborated on the eligibility criteria as follows in lines 80-109: 

• Population: Included: studies targeted at all staff which are/were active within a health care setting (e.g. a whole hospital or specific unit, health center, or community health network) during an outbreak of a coronavirus infection, causing the following diseases: SARS, MERS, COVID-19. Excluded: studies dealing with other infectious disease outbreaks (e.g. ebola and H1N1 virus). 

• Risk or protective factors: Included: studies describing any modifiable risk factor or protective factor, which is relevant to take into account when developing either prevention programs or mental health interventions for HCWs in the context of an infectious disease outbreak. Risk factors are here defined as characteristics at organizational, social and personal level that putatively precede and are associated with poor mental health outcomes in HCWs. Protective factors are positive influences that may protect HCWs for developing mental health problems during coronavirus disease outbreaks. Modifiable factors include behaviors, experiences and exposures that may be controlled and changed for maximizing resilience of HCWs during and after these crises. Examples of modifiable factors are: direct contact with patients, dissatisfaction with procedures, changes in work demands, being quarantined as HCW, fear of infection, stigma, vulnerability, clear communication of directives, professional support, social support, perceived self-efficacy, sufficient precautionary measures, training in protection… We adhered to the author’s interpretation on the classification of risk/protective factors and outcome measures. Excluded: studies describing non-modifiable factors such as gender, age, family history and professional title and factors associated with impact on personal life. Relevant factors were discussed with content experts. Studies solely dealing with prevalence or incidence rates of mental health outcomes are excluded since no data on risk or protective factors can be extracted. 

• Outcome: Included: any mental health outcome or psychological wellbeing. Mental health problems can present within a broad range of severity. Hence, we not only focused on mental health outcomes reflecting psychiatric symptoms and/or caseness, but also included studies assessing mental health symptomatology reflecting common emotional and social undesirable changes in thinking, feeling and/or behavior. Mental health outcomes included physical symptoms such as pain and fatigue in as far as they were part of mental health questionnaires and thus considered putatively related to psychological suffering. Mental health outcomes were considered in the immediate, short- and long-term as long as association with risk/protective factors directly relevant to the disease outbreak could be examined. Excluded: physical health problems related to the inflammatory responses of the human body such as fever, cough, myalgias, chills, headaches, dyspnea, sore throat, nausea/vomiting and diarrhoea.

3- The literature search is inefficient and under my point of view probably the main reason why relevant hits were lost along the process. Not sure the addition of keywords as “Zika virus’/exp”, “‘Zika fever’/exp”, “hotline:ab,ti” led to the inclusion of many relevant hits. However, demanding studies to include one of the keywords in the third paragraph in the literature search in order to be included, must have limited the detection capacity of relevant hits.

We agree with the reviewer that search terms related to the Zika and Ebola virus might be too broad regarding the PICO question but by including these terms we initially ensured that no studies with related infectious disease outbreaks were missed. Since there was a large body of evidence on coronavirus disease outbreaks, we decided to exclude studies dealing with Ebola or Zika viruses as a source of indirect evidence. Including the above-mentioned terms in the search strategy (using the Boolean ‘OR’ operator) made the search strategy more sensitive (broader) and not more specific, which could not have led to missing relevant records. 

The third paragraph of the search strategy included either mental health outcomes or interventions. By combining multiple relevant and sensitive terms such as, amongst others, “mental health”, “psychosocial support” or “stress” all relevant records should be identified in the databases used. The Cochrane Handbook (Chapter 4) also demands that a set of terms dealing with the interventions is included in the search strategy. Searching for outcomes is optional but the outcome terms are here combined with the Boolean ‘OR’ operator with the intervention terms to increase sensitivity. Based on discussions with a content expert we included terms related to “hotline” in the search strategy since it is a potentially relevant psychological service which is often frequented by people in mental distress for providing relief. Furthermore, there were early reports of such services being specifically set up to support HCWs during the COVID-19 pandemic (e.g., Kang et al., 2020)*. 

The search strategies were co-developed by review authors NDB and EDB who are trained medical/healthcare information specialists and content experts (SB and KVP). By doing so, we ensured that we used a sensitive search strategy for detecting relevant studies. In addition to running these searches in the different databases, we also screened the COVID-19 living evidence map from the Nordic Institute of Public Health, containing all experimental and observational studies conducted in the context of COVID-19. This strategy allowed us to be very complete, and not to miss relevant studies. If certain studies were not included, the reason for this were our eligibility criteria or the search date. 

*Kang L, Li Y, Hu S, Chen M, Yang C, Yang BX, Wang Y, Hu J, Lai J, Ma X, Chen J, Guan L, Wang G, Ma H, Liu Z. The mental health of medical workers in Wuhan, China dealing with the 2019 novel coronavirus. Lancet Psychiatry. 2020, 7(3):e14.

4- I am not convinced by the definition of population regarding exposure to coronavirus. Several studies evaluated HCW, particularly during SARS pandemic, both during outbreak and after the outbreak. It seem like authors considered posttraumatic stress symptoms as well, which may appear months/years after. Also, how was hospital setting defined? Were GPs or HCW working in the community excluded?

We indeed included studies assessing the mental health of health care workers (HCWs) which are/were active during an outbreak of a coronavirus infection (see S5 Table for assessment of recall bias). Furthermore, we included HCWs in all different settings. Mental health outcomes were considered in the immediate, short- and long-term as long as association with risk/protective factors directly relevant to the disease outbreak could be examined (see response to comment 6). Lastly, the risk factor of ‘level of disease exposure’ included, amongst others, direct contact with patients, working in high risk units, high risk of exposure, and working on the frontline (see lines 155-156).

In response, we now have better defined the population’s inclusion criteria in lines 80-82 as follows: “studies targeted at all staff which are/were active within a health care setting (e.g. a whole hospital or specific unit, health center, or community health network) during an outbreak of a coronavirus infection, causing the following diseases: SARS, MERS, COVID-19.” 

5- The definition of a risk or a protective factor is really ambiguous and this is particularly problematic. This way it is not possible to replicate what the authors have done. For instance, looking at the studies in the following systematic review (PMID: 32658823) that were not included in this systematic review, it is difficult to know if it was an strict definition of risk/protective factor (or outcome), an inefficient literature search, or losses of studies during the screening process, which led to the authors missing studies that seem relevant to me.

We followed the author’s interpretation on the classification of risk/protective factors and outcome measures. In response, we now defined protective and risk factors as follows in lines 84-99: “Risk or protective factors: Included: studies describing any modifiable risk factor or protective factor, which is relevant to take into account when developing either prevention programs or mental health interventions for HCWs in the context of an infectious disease outbreak. Risk factors are here defined as characteristics at organizational, social and personal level that putatively precede and are associated with poor mental health outcomes in HCWs. Protective factors are positive influences that may protect HCWs for developing mental health problems during coronavirus disease outbreaks. Modifiable factors include behaviors, experiences and exposures that may be controlled and changed for maximizing resilience of HCWs during and after these crises. Examples of modifiable factors are: direct contact with patients, dissatisfaction with procedures, changes in work demands, being quarantined as HCW, fear of infection, stigma, vulnerability, clear communication of directives, professional support, social support, perceived self-efficacy, sufficient precautionary measures, training in protection… We adhered to the author’s interpretation on the classification of risk/protective factors and outcome measures. Excluded: studies describing non-modifiable factors such as gender, age, family history and professional title and factors associated with impact on personal life. Relevant factors were discussed with content experts.” 

We carefully read the systematic review highlighted by the reviewer. While this review appraised the burden (type and frequency) of physical and mental health outcomes in the current literature, we complementary focused on the risk and protective factors which might be associated with mental health outcomes in HCWs. As a result, studies solely reporting prevalence figures were not included in our rapid systematic review. The exclusion criteria in lines 197-99 now read as follows: “Studies solely dealing with prevalence or incidence rates of mental health outcomes are excluded since no data on risk or protective factors can be extracted.” Moreover, we elaborated on the findings of this review in the introduction section in lines 47-48 as follows: “Another recent meta-analysis revealed that HCWs exposed to SARS/MERS/COVID-19 reported symptoms of e.g. fear, insomnia, psychological distress, burnout and anxiety [13]” Moreover, we added the following sentences to the discussion section in lines 325-327: “Very recent meta-analyses confirmed that HCWs exposed to (SARS, MERS or) COVID-19 suffered from a wide range of mental health problems but no risk or protective factors were identified [10-13].”

6- Further explanation of which outcomes were included and excluded would be advisable. How were physical and mental health outcomes defined and differentiated?

We agree with the reviewer that further explanation is advisable. However, as mental health takes into account physical symptoms without identifiable physical cause, physical and mental health symptoms cannot be entirely differentiated. However, we tried to differentiate these outcomes from physical health as relevantly as possible through focusing on mental health outcome questionnaires. Such questionnaires often include physical symptoms such as pain or fatigue as well (e.g., anxiety, burnout or PTSD-related scales), although always putatively related to psychological suffering.

To address this comment, the outcome’s inclusion and exclusion criteria now read in lines 100-109 as follows: “Outcome: Included: any mental health outcome or psychological wellbeing. Mental health problems can present within a broad range of severity. Hence, we not only focused on mental health outcomes reflecting psychiatric symptoms and/or caseness, but also included studies assessing mental health symptomatology reflecting common emotional and social undesirable changes in thinking, feeling and/or behavior. Mental health outcomes included physical symptoms such as pain and fatigue in as far as they were part of mental health questionnaires and thus considered putatively related to psychological suffering. Mental health outcomes were considered in the immediate, short- and long-term as long as association with risk/protective factors directly relevant to the disease outbreak could be examined. Excluded: Physical health problems related to the inflammatory responses of the human body such as fever, cough, myalgias, chills, headaches, dyspnea, sore throat, nausea/vomiting and diarrhoea.” 

7- Pooling together any risk/protective factor with any mental health outcome to provide quantitative results seems questionable to me. Also, do not understand why the outcomes are too heterogenous for a meta-analysis, but not for vote counting.

According to the Cochrane Handbook (Chapter 12), legitimate reasons why it may not be possible to undertake a meta-analysis include amongst others different effect measures used across studies and methodological diversity. In our case, studies reported mean difference, adjusted odds ratio, or regression coefficient as effect measure and it was not possible to pool these different measures. Moreover, the risk/protective factors were evaluated with different scales and it is, hence, not legitimate to perform meta-analyses. Vote counting is based on the direction of effect and can be used according to the Cochrane Handbook when there is inconsistency in the effect measures or data reported across studies, as is the case for this rapid review. Vote counting only addresses the question: Is there any evidence of effect? It does not provide any information on the magnitude of effects and does not account for differences in the relative sizes of the studies. This decision was motivated in the abstract in lines 9-10: “Meta-analysis was not possible because of high variation in risk factors, outcomes and effect measures” and in the method section in lines 124-127: “Since meta-analysis was not possible because of a high variety in factors, outcomes and effect measures (adjusted for different confounding factors), we synthesized the data where possible using vote counting based on direction of effect by comparing the number of comparisons showing harm and benefit, regardless of the statistical significance or size of their results [21].” 

We now also further acknowledged the underlying limitations of this method in the discussion section in lines 308-318 as follows: “Third, since there was a high variation in effect measures, risk factors, and measurement tools, it was difficult to synthesize the findings, and we used vote counting based on direction of effect as synthesis method. As a consequence, statistical heterogeneity could not be assessed. To avoid fragmentizing the body of evidence, we combined several mental health outcomes in this analysis, for the different categories of risk/protective factors. Vote counting provides no information on magnitude of effect and takes no account of the differences in relative sizes of each study. Since the mental health outcomes were grouped and the risk and protective factors were clustered in ten thematic categories, this synthesis method is hence not a nuanced approach. Indeed, these thematic categories (in)directly involve multiple affiliated factors and, by way of example, witnessing multiple deaths could be related to ‘level of disease exposure’ and risk of transmission to family was presently categorized under ‘risk perception and fear’. Nonetheless the results of the individual studies were tabulated and extensively discussed”. 

In addition to this way of summarizing the data, we also tabulated the results and quality of the individual studies in the supporting information (S4 Table and S5 Table), as well as extensively elaborated on these results in the result section, in order to also transparently report and present individual results to the reader.

8- I do not agree with the decision of considering p-value < 0.10 significant regardless of the method and this decision should be better justified.

To address this comment, we decided to consider p-value < 0.05 significant for the results of vote counting. The sentence in line 128 now reads as follows: “A p-value < 0.05 was considered statistically significant.” The discussion of the results on being quarantined as HCW were rewritten accordingly.

9- Including only studies in English in this topic is a limitation as many studies come from Asia.

We agree with the reviewer that this is a limitation, which we now acknowledged in the Discussion section. However, 80% of our included studies were conducted in Asia, which we now also mentioned in the Discussion section in lines 285-289 as follows: “First, because of time pressure of making timely relevant results available during the COVID-19 pandemic, no protocol was registered with PROSPERO, study selection and data extraction was done by only one reviewer and only English language studies were included. The latter has resulted in the exclusion of some studies, particularly from Asian countries. However, we were able to detect 33 individual studies against the predefined eligibility criteria of which about 80% were conducted in Asia.”

---

## [Decision Letter · Decision Letter 3]

26 Oct 2020

PONE-D-20-11380R3

Factors affecting mental health of health care workers during coronavirus disease outbreaks (SARS, MERS & COVID-19): a rapid systematic review

PLOS ONE

Dear Dr. de Brier,

Thank you for submitting your manuscript to PLOS ONE. After careful consideration, we feel that it has merit but does not fully meet PLOS ONE’s publication criteria as it currently stands. Therefore, we invite you to submit a revised version of the manuscript that addresses the points raised during the review process.

Please rerun your search as requested by reviewer 2.

We look forward to receiving your revised manuscript.

Kind regards,

Rosemary Frey

Academic Editor

PLOS ONE

Reviewers' comments:

Reviewer's Responses to Questions

**Comments to the Author**

1. If the authors have adequately addressed your comments raised in a previous round of review and you feel that this manuscript is now acceptable for publication, you may indicate that here to bypass the “Comments to the Author” section, enter your conflict of interest statement in the “Confidential to Editor” section, and submit your "Accept" recommendation.

Reviewer #2: All comments have been addressed

Reviewer #3: All comments have been addressed

2. Is the manuscript technically sound, and do the data support the conclusions?

Reviewer #2: Yes

Reviewer #3: Yes

3. Has the statistical analysis been performed appropriately and rigorously? 

Reviewer #2: Yes

Reviewer #3: Yes

4. Have the authors made all data underlying the findings in their manuscript fully available?

Reviewer #2: Yes

Reviewer #3: Yes

5. Is the manuscript presented in an intelligible fashion and written in standard English?

Reviewer #2: Yes

Reviewer #3: Yes

6. Review Comments to the Author

Reviewer #2: I have read the detailed responses to the comments of reviewer#3 and you have endeavoured to reply to all of them in great detail.

The problem of having an up to date review especially when there is a time sensitive outcome (as with your review, where health care workers and administrators want to know your findings and apply them) is difficult.

Your last search was May 28 in a Covid-19 database. Nearly five months (June, July, August, September and now 22 October) have gone by. It would be reasonable for you to run your search strategy for studies after May 28 and simply state if you found any studies at low risk of bias and if they might change your conclusions substantially. If no such studies are found then you can say so. It is likely that studies on COVID-19 will have continued to be published after May 28 but less likely on the other two corona virus epidemics.

Reviewer #3: Thank you for your detailed response to my previous comments.

In my opinion the manuscript has improved a lot

7. PLOS authors have the option to publish the peer review history of their article (what does this mean?). If published, this will include your full peer review and any attached files.

Reviewer #2: **Yes: **Roger E. Thomas

Reviewer #3: No

---

## [Author Response · Author response to Decision Letter 3]

19 Nov 2020

Reply to the remarks of the reviewers of manuscript ID PONE-D-20-11380R3

Dear Dr. Frey

Dear Editor

Thank you for the work done on manuscript ID PONE-D-20-11380R3 entitled "Factors affecting mental health of health care workers during coronavirus disease outbreaks (SARS, MERS & COVID-19): a rapid systematic review" by Niels De Brier, Stijn Stroobants, Philippe Vandekerckhove and Emmy De Buck.

Reviewers commented positively on our manuscript and clearly stated that all comments have been addressed. Nevertheless, reviewer 2 had one final suggestion to improve the manuscript. Below, we detail how we have addressed this final point raised. 

We have revised the manuscript four times profoundly, including two updates of our searches, and hope that the manuscript will now be acceptable for publication given the timely relevance of the review’s conclusions. 

Yours sincerely,

Niels De Brier, corresponding author on behalf of all co-authors. 

 

Reviewer #1

Reviewer #1 had no further comments on the revised manuscript. 

Reviewer #2

I have read the detailed responses to the comments of reviewer#3 and you have endeavoured to reply to all of them in great detail.

We thank the reviewer for his positive comment. 

The problem of having an up to date review especially when there is a time sensitive outcome (as with your review, where health care workers and administrators want to know your findings and apply them) is difficult.

Your last search was May 28 in a Covid-19 database. Nearly five months (June, July, August, September and now 22 October) have gone by. It would be reasonable for you to run your search strategy for studies after May 28 and simply state if you found any studies at low risk of bias and if they might change your conclusions substantially. If no such studies are found then you can say so. It is likely that studies on COVID-19 will have continued to be published after May 28 but less likely on the other two corona virus epidemics.

In response, we reran our search in the NIPH living COVID-19 map (PubMed results supplemented by regular updates with material retrieved by searches performed by organizations such as World Health Organization, WHO and Centers for Disease Control and Prevention, CDC) on October 26 and screened for additional relevant experimental or observational studies. We were not able to identify studies of higher quality than the studies already included, and provided this information in the discussion section. Moreover, to go beyond the comment of the reviewer, we also synthesized the results of the eight newly identified cross-sectional studies meeting the eligibility criteria. It is important to note that these low-quality studies did not reveal any new risk factors. Taken together, we now have added the following paragraph to the discussion section in lines 382-401: 

“Lastly, the multiple uncertainties about the COVID-19 outbreak and the rapidly growing research make it necessary to provide the scientific community with high-quality and timely updates of the relevant evidence. As this review deals with time sensitive results, we screened for additional relevant experimental or observational studies in the NIPH living COVID-19 map during the publication process (search date Oct 26). In this time period, no high-quality studies at low risk of bias which could substantially impact on the review’s conclusions were identified. Eight additional cross-sectional studies, which suffer from coexisting methodological issues, fulfilled the eligibility criteria. These studies did not reveal any new risk factors and further underpin the findings of this review. When briefly discussing the risk factors, studies by Cai et al. [63] and Wankowicz et al. [64] confirmed that the rate of mental health problems is significantly increased in frontline HCWs. Further, there is growing evidence that risk perception and health fear are independent factors for developing mental distress [65,66] while Sarboozi Hoseinabadi et al. [67] could not demonstrate a significant association between fear of COVID-19 infection and burnout symptomatology. It was also confirmed that loneliness because of social isolation as well as job stress are associated with increased mental distress or burn-out symptomatology, respectively [66,67]. Regarding the protective factors, further evidence shows that receiving adequate information and availability of protective measures are significantly associated with reduced severity of mental health problems [66,68,69]. Sarboozi Hoseinabadi et al. [67] could not demonstrate a significant association between social support from family and friends or hospital resources and COVID-19-related burnout. Finally, Huang et al. [70] found that problem-coping style is the common influencing factor for anxiety in nurses. It is of note that the studies of Arpacioglu et al. [65] and García-Fernández et al. [68] included HCWs as well as non-HCWs and results between groups could not be distinguished.”

Reviewer #3

Thank you for your detailed response to my previous comments.

In my opinion the manuscript has improved a lot. 

We thank the reviewer for his/her positive comment.

---

## [Decision Letter · Decision Letter 4]

3 Dec 2020

Factors affecting mental health of health care workers during coronavirus disease outbreaks (SARS, MERS & COVID-19): a rapid systematic review

PONE-D-20-11380R4

Dear Dr. De Brier,

We’re pleased to inform you that your manuscript has been judged scientifically suitable for publication and will be formally accepted for publication once it meets all outstanding technical requirements.

Kind regards,

Rosemary Frey

Academic Editor

PLOS ONE

Additional Editor Comments (optional):

Reviewers' comments:

Reviewer's Responses to Questions

**Comments to the Author**

1. If the authors have adequately addressed your comments raised in a previous round of review and you feel that this manuscript is now acceptable for publication, you may indicate that here to bypass the “Comments to the Author” section, enter your conflict of interest statement in the “Confidential to Editor” section, and submit your "Accept" recommendation.

Reviewer #2: All comments have been addressed

Reviewer #3: All comments have been addressed

2. Is the manuscript technically sound, and do the data support the conclusions?

Reviewer #2: Yes

Reviewer #3: Yes

3. Has the statistical analysis been performed appropriately and rigorously? 

Reviewer #2: Yes

Reviewer #3: Yes

4. Have the authors made all data underlying the findings in their manuscript fully available?

Reviewer #2: Yes

Reviewer #3: Yes

5. Is the manuscript presented in an intelligible fashion and written in standard English?

Reviewer #2: Yes

Reviewer #3: Yes

6. Review Comments to the Author

Reviewer #2: Thanks to the authors for updating the search which now updates the review for the last 6 months. Thanks for the judicious comment on that search. The review reads well and is well performed. It will be interesting to see the response of medical personnel when the vaccine is available and there are delays or inequalities in provision of the vaccine.

Reviewer #3: Thank you for your detailed response to my previous comments.

I do not have any further comments.

7. PLOS authors have the option to publish the peer review history of their article (what does this mean?). If published, this will include your full peer review and any attached files.

Reviewer #2: **Yes: **Roger E. Thomas

Reviewer #3: **Yes: **Gonzalo Salazar de Pablo

---

## [Editor Report · Acceptance letter]

7 Dec 2020

PONE-D-20-11380R4 

Factors affecting mental health of health care workers during coronavirus disease outbreaks (SARS, MERS & COVID-19): a rapid systematic review 

Dear Dr. De Brier:

I'm pleased to inform you that your manuscript has been deemed suitable for publication in PLOS ONE. Congratulations! Your manuscript is now with our production department. 

Kind regards, 

on behalf of

Dr. Rosemary Frey 

Academic Editor

PLOS ONE